

# Effect of mixing structure on the water uptake of mixtures of ammonium sulfate and phthalic acid particles

Weigang Wang[1,4,†], Ting Lei[2,3,†], Andreas Zuend[5], Hang Su[3], Yafang Cheng[2], Yajun Shi[1], Maofa Ge[1,4,6], Mingyuan Liu[1,4]

[1]State Key Laboratory for Structural Chemistry of Unstable and Stable Species, Beijing National

Laboratory for Molecular Sciences (BNLMS), CAS Research/Education Center for Excellence in

Molecular Sciences, Institute of Chemistry, Chinese Academy of Sciences, Beijing 100190, P. R. China

[2]Minerva Research Group, Max Planck Institute for Chemistry, Mainz 55128, Germany

[3]Multiphase Chemistry Department, Max Planck Institute for Chemistry, Mainz 55128, Germany

[4]University of Chinese Academy of Sciences, Beijing 100049, PR China

[5]Department of Atmospheric and Oceanic Sciences, McGill University, Montreal, Quebec, Canada

[6]Center for Excellence in Regional Atmospheric Environment, Institute of Urban Environment, Chinese

Academy of Sciences, Xiamen 361021, PR China

† These authors contributed equally to this work

*Correspondence to*: Weigang Wang (wangwg@iccas.ac.cn) and Maofa Ge (gemaofa@iccas.ac.cn)

**Abstract.** Aerosol mixing state regulates the interactions between water molecules and particles

and thus controls the aerosol activation and hygroscopic growth, which thereby influences the

visibility degradation, cloud formation, and its radiative forcing. Current studies on the mixing

structure effects on aerosol hygroscopicity, however, is few reported. Here we investigated the effect

of phthalic acid (PA) coatings on the hygroscopic behavior of the core-shell mixtures of ammonium

sulfate (AS) with PA using a coating-hygroscopicity tandem differential mobility analyzer (coating-

HTDMA). The slow increase in the hygroscopic growth factor of core-shell particles is observed





with increasing thickness of coating PA prior to the DRH of AS. At RH above 80 %, a decrease in
hygroscopic growth factor of particles occurs as the thickness of PA shell increases, which indicates
that the increase of PA mass fractions leads to a reduction of the overall core-shell particle
hygroscopicity. In addition, the use of the ZSR relation leads to the underestimation for the measured
growth factors of core-shell particles without consideration of the morphological effect of core-shell
particles. For the AS/PA well mixed particles, a shift of deliquescence relative humidity (DRH) of
AS to lower relative humidity (RH) is observed due to the presence of PA in the well-mixed particles.
The predicted hygroscopic growth factor using the ZSR relation is consistent with the measured
hygroscopic growth factor of the well-mixed particles. Moreover, we compared and discussed the
influence of mixing states on the water uptake of AS/PA aerosol particles. It is found that the
hygroscopic growth factor of the core-shell particles is slightly higher than that of the well-mixed
particles with the same mass fractions of PA at RH above 80%. For our observation of AS/PA
particles may contribute to a growing field of knowledge regarding the influence of coating
properties and mixing structure on water uptake.

**1 Introduction**
The ability of aerosol particles to absorb and maintain water molecules, called hygroscopicity, is
one of the most important physico-chemical properties of atmospheric aerosol particles with
profound implications (Shi et al., 2012; Lei et al., 2014, 2018; Gupta et al., 2015; Hodas et al., et
al., 2015; Zawadowicz et al., 2015; Martin et al., 2017). It might determine the phase state (Mu et
al., 2018), size (Peng et al., 2001; Choi et al., 2002), mixing state, optical properties, and chemical
reactivity of atmospheric aerosols exposed to the environment of the different relative humidities



(Heintzenberg et al., 2001; Rudich et al., 2003; Spindler et al., 2007; Abo Riziq et al., 2007, 2008;
Eichler et al., 2008). Moreover, the change of these properties after water absorption on aerosol
particles can strongly affect the cloud formation, aerosol radiative forcing, global climate, and even
human health (Cheng et al., 2008; Reutter et al., 2009; Rose et al., 2011; Stock et al., 2011; Liu et
al., 2012, 2013; Tie et al., 2017). Therefore, the interaction between water molecules and aerosol
particles is crucial for a better understanding of the aerosol-cloud-climate effects in the atmosphere
(Sjogren et al., 2007; Zamora et al., 2011; Jing et al., 2016).
Atmospheric aerosols contain a complex mixture of inorganic and organic compounds in the
different mixing structures, e.g., externally mixed, internally mixed (Ganguly et al., 2006). The
internally mixed aerosol particles may divided into homogeneous and heterogeneous internally
mixed aerosol particles (Lang-Yona et al., 2009), which could, in turn, strongly influence the water
uptake, optical properties, and the cloud condensation nuclei (CCN) ability of the particles (Lesins
et al., 2002; Falkovich et al., 2004; Zhang et al., 2005; Schwarz et al., 2006; Su et al., 2010). Most
of earlier studies on the hygroscopic behavior of multi-components aerosol focus on the well mixed
particles generated from homogeneously internally mixed solutions (Miñambres et al., 2010; Shi et
al., 2014; Gupta et al., 2015; Jing et al., 2016; Lei et al., 2014; 2018) For example, Choi and Chan
(2002) studied on the effects of glycerol, succinic acid, malonic acid, citric acid, and glutaric acid
on the hygroscopic properties of sodium chloride and AS in the well mixed aerosol particles,
respectively using an eletrodynamic balance. They observed that the deliquescence and
efflorescence of sodium chloride and AS were affected by the presence of different organic
components in the mixed aerosol particles. Concerning the hygroscopicity of the heterogeneity of
internally mixed aerosol particles, such as core-shell particles, there are several studies on



investigating their interaction with water molecules (Ciobanu et al., 2009; Song et al., 2012;
Shiraiwa et al., 2013; Hodas et al., 2015; Song et al., 2018). However, to date, few laboratory studies
on the influence of organic coatings on the hygroscopic behavior of seed particles and the difference
of mixing state effects on the hygrosocpicity of aerosol particles (Zhang et al., 2008; Pagels et al.,
2009; Xue et al., 2009; Lang-Yona et al., 2010; Ditas et al., 2018). E.g., A HTDMA study on the
organic coating effects on the hygroscopicity of AS core was studied by Maskey et al. (2014). They
observed a shift of DRH of AS to lower RH for the core-shell particles due to presence of
levoglucosan coatings. They further compared water absorption on AS/levoglucosan core-shell
particles and the AS/succinic acid core-shell particles. They suggests that difference organic
coatings lead to changes in the hygroscopic properties of core-shell particles. Chan et al. (2006)
investigated hygroscopicity of AS coated with different mass fractions of glutaric acid during two
continuous humidification and dehumidification cycles using a Raman spectra and an
electrodynamic balance. They observed different hygroscopic behavior and morphology of aerosol
particles between the two humidification and dehumidification cycles due to the different mixing
states. Therefore, to investigate organic coating effect on the hygroscopicity of seed aerosol particles
and further to study on difference of mixing states effects on the hygroscopic behavior of aerosol
particles are crucial for estimation the direct and indirect radiative effect of aerosol particles on the
Earth's climate (Saxena et al., 1995; Ansari et al., 2000; Maskey et al., 2014).
PA is ubiquitous in rural mountains and marine atmosphere in Asia (Wang et al., 2011). It is mainly
produced by the photo-oxidation of volatile organic compounds (VOCs), such as xylene, and
naphthalene (Kawamura and Ikushima, 1993; Schauer et al., 1996; Zhang et al., 2016). PA has also
been identified as a significant contributor to the urban organic compounds (Rogge et al., 1993). PA





particles are generally used as a tracer for the secondary organic aerosol (SOA) in atmospheric fine
particles (Schauer et al., 2000, 2002). Recently, Zhang et al. (2016) reported the importance of
atmospheric PA aerosol particles in the visibility degradation and the formation of CCN. The organic
PA can have profound effect on light scattering, hygroscopictiy, and phase transition properties of
multicomponent atmospheric aerosols. However, these physico-chemical properties of PA have
been little documented in the literature (Brooks et al., 2004; Liu et al., 2016). Here we summarized
a few studies on the hygroscopicity of the PA-containing aerosol particles. Brooks et al. (2004)
investigated continuous hygroscopic growth of PA aerosol particles in the humidification process
using a HTDMA technique. Hori et al. (2003) and Huff Hartz et al. (2006) measured the high CCN
activity of PA in spite of its low solubility. Also, the liquid-liquid phase separations (LLPS) with
aerosol particles consisting organic and inorganic components were observed by many groups
(Ciobanu et al., 2009; Betram et al., 2011; Song et al. 2012a, 2012b; You et al., 2014). For example,
Song et al. (2012a, 2012b) investigated that LLPS occurs in the mixed dicarboxylic acids containing
5, 6, and 7 carbon using an optical microscopy and micro-Raman spectroscopy, and further
established that occurrence of LLPS of aerosol particles has an average elemental oxygen-to-carbon
(O:C) ratio of the organic fraction of less than 0.8. Subsequently, the occurrence of liquid-liquid
phase separation in the internally mixed aerosols consisting of AS and PA was performed by Zhou
et al. (2014) during the dehumidification processes using the Raman spectra. You et al. (2014)
further found that the LLPS of aerosol particles with a different O: C ratio of 0.5<O: C<0.8 depends
on the types of organic functional groups and inorganic salts presented. Therefore, these studies
suggest that the LLPS in the mixed organic and inorganic components aerosol particles is influenced
by the amounts of organic and inorganic aerosol components and types.





In this work, we investigated the effect of the different thickness of coating PA and the core size on
the water uptake of core-shell particles containing AS, and further studied the effect of the mixing
states on the hygroscopic behavior of PA /AS aerosol particles using a HTDMA technique. For
example, we compared the hygroscopic behavior of well-mixed AS/PA particles with that of core-
shell AS/PA particles with the same PA mass fractions. In addition, we used the Zdanovskii-Stokes-
Robinson (ZSR) relation and the Aerosol Inorganic-Organic Mixtures Functional groups Activity
Coefficients (AIOMFAC) model (Zuend et al., 2008; 2011) to predict the hygroscopic growth factor
(GF) of mixed aerosol particles in the different mixing structure. Moreover, the AIOMFAC-based
model with a version of the liquid-liquid equilibrium (LLE) algorithm was employed in our study
to predict the phase compositions of liquid and solid phases for a given composition of a mixture
(Zuend and Seinfeld. 2013).

**2 Experimental and modeling methods**
**2.1 HTDMA setup and experimental protocol**
A HTDMA setup is employed to measure the aerosol nanoparticle hygroscopic growth factor ($g_f$)
and phase transition in the RH range from 5% to 90 %. Here, $g_f$ is defined as the ratio of mobility
diameter of aerosol particles after humidification ($D_m$(RH)) to that dry condition ($D_m$(<5% RH)).
Figure 1 shows a schematic diagram of the HTDMA setup. It is comprised of four main components,
including three differential mobility analyzers (DMA), a condensation particle counter (CPC), a
humidification system, and a coating system. The more detail information on the HTDMA setup,
calibration, and verification have been described elsewhere (Lei et al., 2014; 2018; Jing et al., 2016;
Liu et al., 2016). In our study, the particle-sizing, the aerosol/sheath flow rates, and DMA voltage



supply have been calibrated every month, respectively. The uncertainty of aerosol/sheath flow rates
are kept within ±1% around the reference values. The deviations of the measured DMAs voltage
from set-point values is less than ±1%. The sizing agreement of DMAs between measured diameter
of polystyrene latex (PSL) spheres and their nominal diameter (100±3 nm) is within ±1%. The
chemical substances and related to physical properties are available in the Supporting Information
Table S1. The solutions used in our measurements are prepared with distilled and de-ionized millo-
Q water (resistivity of 18.2 MΩ cm at 298 K).
**2.1.1 Homogeneously internally mixed AS/PA Aerosol particles**
Briefly, poly-disperse aerosol particles were atomized from homogeneous bulk solutions with
different mass fractions of PA and AS (Fig. 1), assuming that the compositions of aerosol particles
remain the same as that of the solutions used in the atomizer (MSP 1500, MSP). The resulting
particles were dried and subsequently charged through a dryer and then a neutralizer, respectively.
A mono-disperse distribution of particles with a desired diameter were selected by the first
differential mobility analyzer (DMA1) with RH below 5 %. After particle sizing, the aerosol
particles were exposed to a humidification mode (5%→90%) in the Nafion conditioner tubes. The
number size distributions of humidified aerosol particles were then measured by a DMA3 coupled
with a CPC. To have a precise control of the aerosol RH, the flow rates of the humid and dry air
were adjusted with a proportional-integral-derivative (PID) system. Also, to ensure the sufficient
water equilibrium with aerosol particles, the difference between RH2 and RH3 (RH in the sheath
flow) was within 2 % during the experiment.
**2.1.2 Heterogeneously internally mixed AS/PA aerosol aerosols**
The seed aerosol particles were generated from an aqueous solution of AS (0.05 wt%) by an atomizer.




After a passage through a silica gel diffusion dryer and a neutralizer, the seed aerosol particles with
a certain diameter (100, 150, 200nm, respectively) were firstly selected by a DMA1 and then
exposed to organic vapors in a coating system. To be specific, the coating system contains a
controlled silicone oil bath vaporizer, a reservoir of organic compound, and a condenser. The seed
AS particles passed through a sealed flask immersed in a silicone oil bath. The sealed flask was
filled with the PA powder. The PA vapors were enriched into the aerosol flow by heating. The
resulting organic vapors were condensed onto the seed particles after cooling to an ambient
environment through a condenser. Similarly, this system for coating organic components on the
particles has been proved to be efficient by Abo Riziq et al. (2008). The coated particles of certain
sizes were then selected by the DMA2 to determine the thickness of organic components ($D_{total} =$
$D_{core} +$ coating). After core-shell particle-sizing, aerosols pre-humidified in a Nafion tube and flowed
into the second Nafion humidifier at the set RH2 to reach equilibrium at the RH condition. Finally,
the conditioning core-shell particle was detected by a DMA3 and a CPC at ambient temperature.
The uncertainty of thickness of coating PA was ± 1.0 nm, considering the fluctuation in temperature
and uncertainty of sizing measurements by DMAs.
**2.2 Theory and modeling methods**
**2.2.1 GF data fit**
We use the following expression to predict the hygroscopic growth factor of individual components.
$$GF = \left[1 + (a + b * a_w + c * a_w^2) \frac{a_w}{1-a_w}\right]^{\frac{1}{3}}$$    (1)
Here it is assumed that water activity ($a_w$) is equal to the water saturation ratio ($a_w$ = RH / 100 %).
The coefficients a, b, and c are determined by fitting Eq. (1), and their values are shown in Table 1
according to the measured GF data against RH. The equation (1) is expected to fit the continuous



water uptake behavior of particles (Brooks et al., 2004; Kreidenweis et al., 2015).
**2.2.2 GF predictions by ZSR**
We use the Zdanovskii-Stokes-Robinson (ZSR) relation to calculate the hygroscopic growth factor
of mixed particles, $GF_{mixed}$. The $GF$ of a mixture can be estimated from the $GF_j$ of the pure
components $j$ and their respective volume fractions, $\mathcal{E}_j$, in the mixture (Malm and Kreidenweis,

183    1997).

$$GF_{mix} = \left[ \sum_j \varepsilon_j \, (GF_j)^3 \right]^{\frac{1}{3}}$$    (2)
$$\varepsilon_{AS} = \frac{\frac{4}{3}\pi R_{AS}^3}{\frac{4}{3}\pi R_{core-shell}^3}$$    (3)

**3 Results and discussion**
**3.1 Hygroscopic growth of homogeneously internally mixed aerosol particles**
Figure 2 shows the measured hygroscopic growth factors of the AS, PA, and well-mixed AS with
different mass fractions of PA particles with dry diameter 100nm against RH, respectively. During
the hydration mode, there is no change in size until a slow increase occurs at 60% RH in the case of
well-mixed AS/PA aerosol particles. This increase may occur because the PA uptakes a small amount
of water. However, an abrupt increase in the hygroscopic growth factor is observed at 75% RH for
well-mixed particles containing 50wt, 75wt % PA, of which the growth factor is higher than that of
pure PA aerosol particles (1.09 ± 0.01 nm from measurements shown in Fig. 2) at the same RH. An
interesting, yet contrasting phenomenon is that water uptake for well-mixed particles containing
50wt % PA components is relatively higher than that of mixtures containing 75wt % PA at 75% RH.
One possible reason is that the full deliquescence of AS in the well-mixed particles with 50wt % PA
components is completed at 75 % RH, while AS in the mixtures containing 75wt % PA components



is partially deliquescent. A decrease in the hygroscopic growth factor of well-mixed AS/PA particles
with increasing mass fractions of PA is observed at RH above 80 %. For example, the measured
growth factors for internal mixtures containing 25wt, 50wt, 75wt % PA are 1.36, 1.28, 1.19 at 80%
RH, respectively, lower than the growth factor of 1.45 for pure deliquesced AS particles (value from
measurements shown in Fig. 2) at the same RH. Also, the measured hygroscopic growth factors
within experimental uncertainty were in good agreement with the results from the well-mixed
particles performed by Hämeri et al. (2002). In addition, with increasing mass fractions of PA in the
well-mixed particles, the smoothing of the hygroscopic growth factor curves is obvious, indicating
that the PA aerosol particles have a significant effect on the water uptake of well-mixed AS/PA
particles such as a shift or suppression of DRH of AS in the mixed particles. For example, in the
case of 1:3 mixtures of AS:PA (by mass), 75wt% PA in the well-mixed particles suppresses the
deliquescence of AS, i.e., AS in the well-mixed particles slowly dissolve into the liquid phase due
to continuous water uptake of PA prior to the deliquescence relative humidity of AS (80% RH). This
similar phenomenon was observed by previous studies (e.g., Hämeri et al., 2002; Qiu and Zhang,
2013). For example, Qiu and Zhang (2013) observed that mixture particles consisting of
dimethylaminium sulfate and AS exhibited a moderate growth by water uptake in the RH range of
40%-70% RH. The calculated growth factors from a model based on the ZSR relation agree well
with the hygroscopic growth factors of well-mixed AS/PA particles when accounting for
measurement uncertainty. A possible reason for this good agreement is that the measured growth
factors referring to the water uptake contribution by PA in the ZSR relation are obtained from the
fitted growth curve of pure PA particles (as shown in Fig. 2. Fitted expression, Eq. (1)). Thus, in the
case of well-mixed AS/PA, relatively good agreement with the experimental growth factors of





mixtures with 25, 50, and 75 wt % PA demonstrates that individual components independently
absorb water in proportion to their volume. However, the discrepancy between measured growth
factor of well-mixed AS/PA particles at 75 % RH and the predicted growth factors by using the ZSR
relation may be due to the molecule interaction between organic molecular and completely or
partially dissolved AS ions. A similar phenomenon was reported for well-mixed mixtures of AS +
levoglucosan in the previous study by Lei et al (2014, 2018).
**3.2 Hygroscopic growth of core-shell structured aerosol particles**
Figure 3 shows the measured hygroscopic growth of core-shell structured particles as a function of
RH. Here, we investigated the hygroscopic behavior of samples of various seed particle sizes (AS
particle dry diameter of 100, 150, 200nm) and coating (PA coating of 10, 20, 30, 50nm), respectively.
The core-shell structured particles start to absorb a small amount of water at RH lower than the
DRH of AS due to the organic coating. A similar behavior has been observed for core-shell
structured particles containing AS and palmitic acid by Garland et al. (2005), where early water
uptake and reduced hygroscopic growth after deliquescence of AS (compared to pure AS aerosols)
were reported. A reduction of the hygroscopic growth factors of core-shell particles becomes
obvious as the thickness of the PA shell increases after the deliquescence of core-shell particles. For
example, the measured growth factor value at 80% RH is 1.45, 1.40, 1.32, and 1.28 for core-shell
particles containing 100nm AS and 10, 20, 30, 50nm coating PA shell, respectively. The kinetic
limitation on the core-shell particles is expected to increase considerably with increasing the
thickness of the coating PA shells, which retards the transport rate of water molecules across core-
shell aerosol particles/air interface. In addition, the measured hygroscopic growth factor of core-
shell AS/PA mixtures is predicted by the ZSR relation, which is based on the hygroscopic growth



factors of AS and PA derived from the E-AIM predictions for AS and the fitted GF curve (Eq. 1).
The ZSR-based predictions are lower than that of core-shell aerosol particles at RH in the range of
5-90%. The underprediction of the ZSR relation was also observed in the literatures (Chan et al.,
2006; Sjogren et al., 2007). Sjogren et al. (2006) observed a strong higher water uptake of mixtures
consisting of AS and adipic acid with different mass rations (1:2, 1:3, and 1:4) at RH above 80 %
compared with ZSR relation in the hydration condition. They assumed that adipic acid is more likely
to enclose the water-soluble AS in veins and cavities, which results in easy uptake of water and a
negative curvature of the solution meniscus at the opening of the vein compared to a flat or convex
particle surface. Thence, in the case of AS/PA core-shell particles, one potential reason for the
underestimation of the measured growth factor by ZSR relation is the morphology effect on the
core-shell structured AS/PA particle. To be specific, for the core-shell aerosol particles consisting of
PA and AS, especially at 80% RH, it shows a considerable amount of water uptake due to the
dissolution of the AS core. This dissolution of AS may form completely or partially mixed AS/PA
solution droplets. The resulting effect of the arrangement and restructuring of core-shell structured
particles may change the hygroscopicity and mixing state of the core-shell particles at RH above
80% (Chan et al., 2006; Sjogren et al., 2007). Another morphological effect could be that
morphology of a somewhat porous polycrystalline AS core could lead to a larger amount of AS in
the particles at RH prior to deliquescence of AS – to appear as a 100-200 nm mobility diameter –
hence a thinner than 10-50 PA coating to bring it to a near spherical shape of 110-250 nm core-shell
particles (Zelenyuk et al., 2006).
Figure 4 shows that the experimental water absorption of the varying size of AS core coated with
50-nm PA shell in the hydration condition. In the case of 50nm-PA shell coated with a certain size



of the AS seed (100, 150, 200nm) with respect to 68wt, 55wt, 46 wt % PA in the core-shell particles,
It exhibits an increase in hygroscopic growth factor of core-shell particles at RH below 80 % as the
size of AS core decreases. However, a decrease in hygroscopic growth factor of core-shell mixtures
is observed at RH above 80 % with decreasing the size of the AS core. This indicates that the 50nm-
PA shell in the core-shell particles have predominantly contributed to the hygroscopic growth of
core-shell particles at low RH. At high RH (e.g., after AS deliquescence), however, 50-nm PA
coating shows a weak kinetic limitations for water uptake by core-shell particles as the size of AS
core increases. For example, the measured growth factor value is 1.28, 1.34, and 1.40 at 80% RH
for 100-200 nm AS core in the mixed particles, respectively. The discrepancy between measured
hygroscopic growth factors and predicted hygroscopic growth factors of core-shell particles by ZSR
relation, as discussed in Sect. 3.2, is due to the morphology effect. For ZSR prediction, it assumes
volume fraction of AS components is constant according to the ratio of volume of AS core in the
sphere to the volume of core-shell sphere based on Eq. (3). Without considering morphology effect,
the ZSR prediction results in an underestimation of hygroscopic growth factors of core-shell
particles.
**3.3 Comparison of heterogeneously and homogeneously internally mixed AS/PA aerosol**
**particles**
Figure 5 shows the hydration curves of different AS cores coated with the different mass fractions
of PA loading (shown in the Supporting Information Table S2) in comparison with those of the well-
mixed with the same PA mass fractions particles, pure AS particles, and pure PA particles in the
range of 5 − 90% RH. The effect of the coating PA on core-shell particles becomes more pronounced
than that of PA in the well-mixed particles at RH below 70% as shown in Fig 5a-b, which leads to



higher amounts of water absorption at low RH. However, compared to Fig 5a-b, Fig. 5c shows the
hygroscopic growth factors of well mixed AS/PA is slightly higher than that of AS/PA core-shell
particles with 46% wt PA. At 75% RH, the measured growth factor value of core-shell particles is
lower than that of homogeneously internally mixed mixtures in the PA mass fraction range from
68wt to 46wt % due to the mass transfer limitations of water vapor transport to the AS core in the
core-shell particles. For the well mixed AS/PA particles, however, partial dissolution of AS into the
liquid AP phase may lead to more water uptake by well mixed particles. For example, for the core-
shell mixtures with 68wt % PA loading, the experimental growth factor value is 1.09 at 75% RH,
relative to the growth factor of 1.17 of well-mixed mixtures AS/PA. After an abrupt increase in
particle diameter of mixed particles, the core-shell AS/PA particles uptake slightly more water than
well-mixed AS/PA with the same mass fractions of PA as RH increases above 80%. Core-shell
particle morphology may experience the restructuring and associate size change of particles. A
similar hygroscopic behavior was observed in previous papers (Chan et al., 2006; Sjogren et al.,
2007; Maskey et al., 2014). Chan et al. (2006) observed for hygroscopicity of 49wt % glutaric acid
coated on AS core during two continuous hydration cycles: the experimental growth factor of the
fresh core-shell of AS and glutaric acid in the first hydration cycle is a bit higher than those in second
hydration cycle with the same mass fractions of glutaric acid. They suggested that the mixing state
has changed from core-shell to well-mixed state during the humidification process. Also, a slightly
higher growth factor of core-shell particles than that of well-mixed particles was found when
comparing the hygroscopic growth factors of 49wt % glutaric acid coated on AS core with that of
well-mixed mixtures of AS with the same mass fractions of glutaric acid from different papers (Choi
et al., 2002; Chan et al., 2006). However, a contrasting observation was observed in the previous





study (Maskey et al., 2014). Maskey et al. (2014) investigated the hygroscopic behavior of the
internal mixtures consisting AS coated with either succinic acid or levoglucosan in the different
mixing state with the same volume fractions of organic compounds. The growth factor of core-shell
particles consisting of AS and succinic acid is lower than that of the well-mixed particles, while
experimental values for core-shell of AS/levoglucosan particles are close to those of the well-mixed
mixtures. The possible reasons for the difference between our study and results from Maskey et al.
(2014) are physical properties of the organic components, such as hygrosocpicity, viscosity,
volatility, gas/liquid/solid diffusion coefficient of water vapor, and water uptake coefficients.
Thence, different kinds of organic compounds have a different effect in the hygroscopic growth of
mixtures, including the core-shell and the well-mixed state. For example, no hygroscopic growth
was observed up to 99 % RH for pure succinic acid particles (shown in Fig. S1a). Peng et al. (2001)
measured the DRH of succinic acid at 99% RH using a bulk solution at 24 °C. Also, Henning et al.
(2002) observed no hygroscopic growth factors of soot/succinic acid core-shell particles in the
hydration mode using a HTDMA. In the case of AS/succinic acid core-shell particles, No water
uptake by AS coated succinic acid shell was observed before 80% RH, while there is a gradual
increase in water absorption of core-shell particles prior to the deliquescence of AS with different
mass fraction of PA components as shown in Fig 5a-c. This suggested the physical state of shell is
solid and liquid for Maskey et al. (2014) and our measurements, respectively. At RH above 80%,
the kinetic limitation on the water vapor uptake through solid shell into the core is more obvious
than that through liquid shell into the core (i.e., liquid diffusion coefficient of water vapor is the
range of $10^{-10}$ to $10^{-9}$, solid diffusion coefficient of water vapor is the range of $10^{-13}$-$10^{-14}$ at 25°C).
This can lead to different hygroscopic behavior of core-shell particles. In the case of





AS/levoglucosan measured by Maskey et al. (2014), they found that the slightly higher growth
factors for the well-mixed particles is than core-shell aerosol particles (88nm AS core coating 12nm
levoglucosan). The mass fraction of levoglucosan in the core-shell particles is ~ 29wt %. In our
study, AS coated with PA with mass fraction range is between 46wt to 68wt %. Using the lower
mass fraction of PA (e.g., 29wt %), compared to mass fraction range of 46-68wt %, it may occur
lower hygroscopic growth factors of AS/PA core-shell particles than that of well mixed particles.
The low mass fraction of PA will be explored in future. In addition, by using the ZSR relation to
predict the hygroscopic growth factors of the internal mixture in the different mixing structures with
the same mass fractions of organic compounds, the measured growth factors of well-mixed AS/PA
particles agree well with the calculated growth factors by the ZSR relation comparing with that of
core-shell AS/PA particles.

**4 Summary and conclusion**
Due to different sources and aging process of aerosol particles in the atmosphere, atmospheric
aerosol particles tend to exist in different mixing structures, such as externally mixed, homogeneous
mixed (e.g., well-mixed and core-shell structure). In this work, PA is used as a representative organic
component generated from various sources, such as vehicles, biomass burning, photo-oxidation to
investigate hygroscopic behavior of AS/PA aerosol particles with different mixing states.
Continuous water absorption by pure PA aerosol particles has an important contribution in
smoothing of hygroscopic curve of AS/PA well-mixed mixtures with increasing mass fraction of PA
components. In addition, the ZSR relation is a good estimation of experimental hygroscopic growth
factors of AS/PA well-mixed particles. Furthermore, A coating-HTDMA technique study on the PA



coating effects on the hygroscopicity of AS core is investigated. PA coating increase the water
uptake by core-shell at RH prior to the AS deliquescence but decrease hygroscopic growth of core-
shell particles at high RH. Finally, we and compared and discussed the difference of influence of
mixing structures on the hygroscopicity of AS/PA aerosol particles. In addition, in this study, using
an E-AIM model, Fitted Expression Eq. (1), and the ZSR relation is to predict the measured
hygroscopic growth factors of pure components, well-mixed, and core-shell particles, respectively.
According to filed studies reported in the previous literature, a variety of organic aerosol particles
were characterized in the atmosphere. Thence, the effect of various organic substances on the
hygroscopic behavior cycle of the organic/inorganic core-shell particles need to be further
investigated. Currently, suppression or delay of the DRH and ERH of core-shell particles to some
extent depends on the different types of organic compound coatings, such as molecular structure,
viscosity, solubility and hygroscopicity. Also, for the certain organic compounds such as PA, a
difference in the hygroscopic behavior of mixing states is more likely to depend on the difference
of influence of kinetic limitations. Moreover, in order to understand which models suitable to
explain these potential atmospheric relevant core-shell aerosol hygroscopicities, and whether they
contain any rules related to functional groups of the organic components, it is significant to explore
the possibility of modeling combining with the experimental measurements. Understanding the
contribution of different organics coating to hygroscopic behavior of the core-shell mixtures as well
as discussion of the extent to which kinetic limitations or organic physico-chemical properties is
expected to have contributed to difference in the hygroscopicity in the different mixing structure,
which may lead toward a more mechanistic understanding of how water uptake can be linked to the
mixing states in the atmosphere.

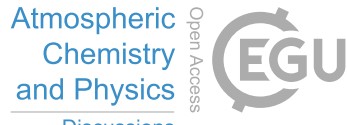
This work focus on the water uptake by these aerosol particles in the different mixing states.
Atmospheric aerosol particles may undergo the humidity cycles depend on the ambient RH history.
A hysteresis effect with solid-liquid phase transition of core-shell aerosol particles may occur as RH
decreases in the ambient air. Also, more attention to the residence time will be paid on core-shell
aerosol particles to reach equilibrium in the whole of RH range. These changes related to the
hygroscopic behavior of core-shell aerosol particles studies will be a topic in the future.

**Author contributions**: W.G.W designed and led the study. W.G.W and T.L assembled the
coating-HTDMA. T.L performed the experiments and prepared the manuscript with contributions
from all co-authors. All co-authors discussed the results and commented on manuscript.
**Data availability**
The data used in this study are available upon request from the corresponding author.
**Acknowledgement**
This project was supported by The National Key Research and Development Program of China
(2017YFC0209500), and the National Natural Science Foundation of China (41822703, 91744204,

391   91844301).

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






**Tables**

**Table 1.** Coefficients of the fitted growth curve parameterization to measured growth

factor data using Eq. (1)

| Chemical Compound | a | b | c |
|---|---|---|---|
| Phthalic Acid | 0.083116 | 0.291473 | -0.353544 |




















**Figures**


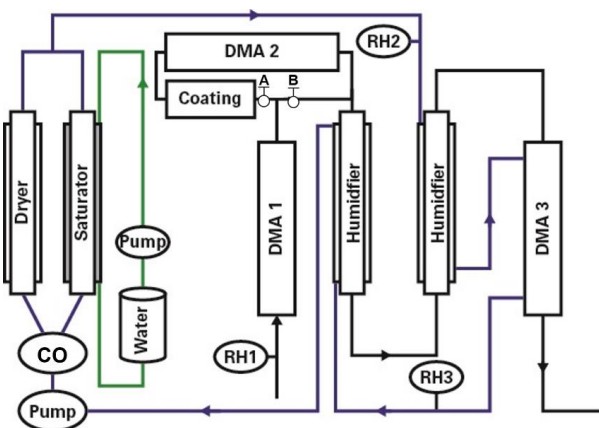


**Figure 1.** Schematic diagram of the coating-hygroscopicity tandem differential mobility analyzer. Here,

CO: critical orifice; DMA: differential mobility analyzer; RH1 and RH2 (measured RH sensor) represent

the RH of aerosol and humidified flow in the inlet of DMA1 and humidifier, respectively. RH3 (measured

by dew point mirror) represent the RH of excess air. Valve B is open and valve A is closed to the

homogeneous internally mixed-mode experiment. Valve A is open and Valve B is closed to the coating-

mode experiment. Black line: aerosol line; Blue line: sheath line; Green line: MilliQ water.









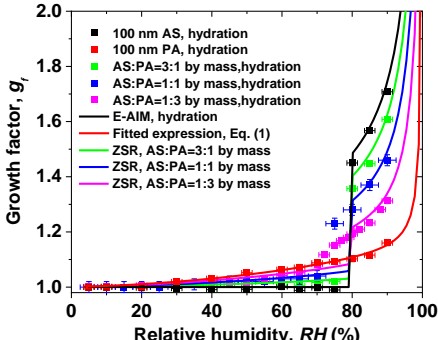


**Figure 2.** Hygroscopic growth factor for 100 nm (dry diameter, RH < 5 %) aerosol particles containing:

ammonium sulfate (AS), phthalic acid (PA), and well-mixed mixtures of PA and AS with different mass

ratio of AS to PA. In comparison, the E-AIM model, the Fitted expression Eq. (1), and the ZSR relation

predicted growth factors of AS, PA, and well mixed particles with different mass fractions of PA,

respectively.











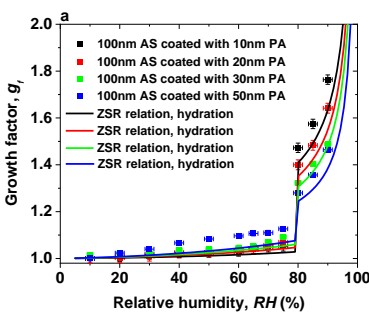


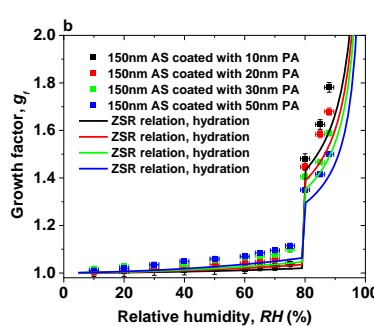


**Figure 3.** Hygroscopic growth factor for core-shell of ammonium sulfate (AS) and phthalic acid (PA)

aerosol particles. In comparison, the ZSR relation predicted growth factor of core-shell aerosol particles

(**a**) 100-nm AS core (**b**) 150-nm AS core (**c**) 200-nm AS core.







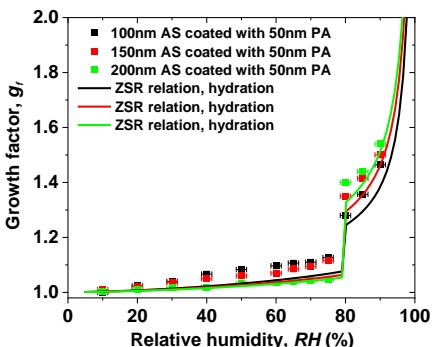


**Figure 4.** Hygroscopic growth factor for 100-200 nm ammonium sulfate (AS) core with coating 50 nm

phthalic acid (PA). In comparison, the ZSR relation predicted growth factor of core-shell aerosol particles

with different AS cores.














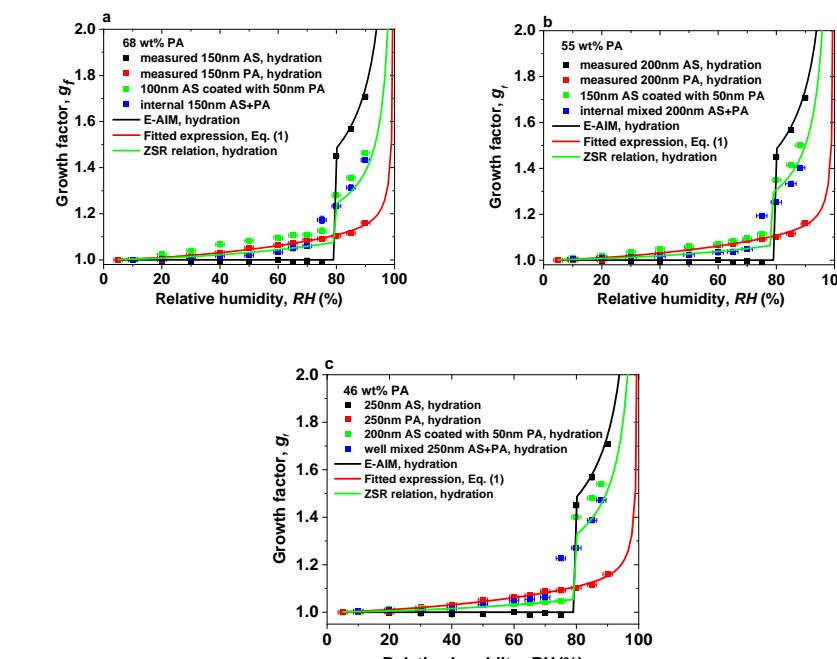




**Figure 5.** Hygroscopic growth factor for core-shell and well-mixed aerosol particles with the same dry

mass fractions of phthalic acid (PA) containing: (a): 68 wt % PA, (b): 55 wt % PA, (c): 46 wt % PA. In

comparison, the E-AIM model, the Fitted expression Eq. (1), and the ZSR relation predicted growth

factors of ammonium sulfate (AS), PA, and internally mixed particles with different mass fractions of

PA, respectively.




