# Peer review of "Effect of mixing structure on the water uptake of mixtures of"

_Atmospheric Chemistry and Physics, 2020_

## Referee Comment (RC1) · Anonymous Referee #1 · 3 Aug 2020

Wang et al investigated the effect of mixing structure on the water uptake of mixtures of ammonium sulfate and phthalic acid particles by taking homogeneously internal mixture and core-shell structure into account. The aerosol water uptake is an important factor to determine many atmospheric processes, such as multiphase reactions and visibility reduction. The hygroscopicity of inorganic and organic mixtures is not well-understood. The authors emphasized the importance of the particle morphology and mixing structure for inorganic-organic mixture water uptake. I would like to recommend it to publish to ACP after some minor revisions.

Comments: 1. Regarding the core-shell structure, the particle morphology may change

when the particles are exposed to a high relative humidity condition. This could attribute to both core and shell take up enough water and get mixed together. Thus, the core-shell structure could be ambiguous. 2. The kinetic limitation has been referred for explaining the effect of core-shell structure on particle water uptake. In HTDMA system, the residence time of particles passing through the conditioned part may be short and lead to a kinetic limitation. This could not be the case when RH exceeds the DRH and particles are liquefied. We should also note that the particles with core-shell structure may have enough time to eliminate such kinetic limitation in the real atmosphere, different from the situation in the HTDMA system. 3. In addition, if the phthalic acid is surface active? The reduction in surface tension may be closely related to the hygroscopicity of core-shell particles. 4. There are numerous English grammar errors, especially in the summary and conclusion section (Line 354-360).

---

## Referee Comment (RC2) · Anonymous Referee #2 · 31 Aug 2020

Wang and Lei et al., present a study about the hygroscopicity of aerosol particles consisting of ammonium sulfate (AS) and phthalic acid (PA). Using a HTDMA setup, the authors study first the hygroscopicity of particles where AS and PA are present in a well-mixed, internal mixture and then particles consisting of AS core and PA shell of varying size. Later, the authors compare the hygroscopicity of internally mixed particles to those with core-shell structure.

The authors show that at RH above 80% the core-shell particles have higher hygrocopicity than the well-mixed particles. Further, a traditional ZSR-relation, coupled with an empirical growth factor fit for PA hydration curve, predicts a lower hygroscopicity

than what is measured for the core-shell particles. These differences in predicted and measured hygroscopicity are attributed to particle morphology changes, i.e., the shape of the particles deviate from a spherical shape.

The manuscript is generally well written and the results increase the understanding of Atmospheric Chemistry and Physics community about the hygroscopicity of complex organic-inorganic particles. I recommend that the manuscript is published after a minor revision.

General comments

1. How the uncertainty of the measurements was determined? In Figures 2-5 the measured points have uncertainty both in the RH and growth factor direction. However, in the manuscript no information is given how this uncertainty was calculated.

Specific comments

1. Section 2.2.2. Please explain what the different R symbols are in Eq. (2) and (3). Supposedly they refer to radii of spheres.

2. Lines 242–244. To me it looks like the ZSR relation does not predict the hygroscopic growth of AS/PA core-shell particles. The sentence starting from line 245 states the same ("The ZSR-based predictions are lower than...")

3. Lines 338–342. I do not understand this sentence. Please rephrase it. Do you mean that in the future you will explore why the ZSR relation predicts lower growth factors than what is measured for the core-shell particles?

4. The manuscript contains several typographical or grammar errors.
* * *

---

## Referee Comment (RC3) · Anonymous Referee #3 · 31 Aug 2020

Review of "Effect of mixing structure on the water uptake of mixtures of ammonium sulfate and phthalic acid particles" by Weigang Wang et al.

**Summary:**

Wang et al. investigated the effect of different internal mixing structures (homogeneously mixed and core-shell structure) on the water uptake of aerosols consisting of ammonium sulfate (AS) and phthalic acid (PA). In addition, they studied how the amount of PA in the particle affect the water uptake. They used specific HTDMA-instrument to select specific sized particles, add PA coating into them (when core-shell structures were studied) and humidify then following by measurement of the growth of the particles as they uptake water. To accompany the measured data, they used theory for estimating the hygroscopic growth of individual components and Zdanovskii-Stokes-Robinson (ZRS) relation to calculate the hygroscopic growth of mixed particles. For homogeneously mixed particles (also referred as well-mixed particles) they observed, for example, that a decrease in the hygroscopic growth factor (GF) with increasing mass fractions of PA at above RH 80% level. They state that these results also agreed with previous studies, and the predictions from ZRS also agreed rather well for the well-mixed particles. For the core-shell structured particles, Wang et al. observed an increase in the GF as the size of the AS core decreased (below 80% RH level). At above 80% RH level, they observed a decrease in the GF with decreasing size of the AS core. For the core-shell structured particles ZRS predictions underestimated the hygroscopic growth. As a general comment, the methods and experimental procedures are adequately described, and they seem valid for this type of study. My main criticism concentrates on to the relevance of the study, and what new information it brings to the scientific community.

However, I do recommend this paper for publication if the issues raised below are adequately addressed.

**Major comments:**

Why liquid well mixed AS-PA would have different hygroscopicity compared to the AS particle with PA coating? Or was this the research question of the study?

The introduction of the draft is strongly focused on the water uptake of the aerosols which is the main theme of the paper. The atmospheric relevance of phthalic acid is discussed in the introduction (lines 86-98) into some extent, and shortly mentioned in the conclusions (lines 347-349). However, as published studies about the hygroscopicity of organic coatings with inorganic core do exists could you provide more explanation what new this study brings and how PA it its relevant to the atmosphere in larger scale and thereby justify its use in this specific study? Can the results acquired for PA be generalised for other organic compounds too? For example, the authors mention that PA is common in rural mountains and marine atmospheres in Asia, so is it specific to those areas only? Could you, for example, give an estimate how much of PA there is in the atmosphere compared to other organic compounds? In addition, even though the use of ammonium sulfate is very common for this type of studies, its relevance/why it was used could also be shortly mentioned.

As you note in the introduction (lines ), with $0.5 < O:C < 0.8$ the LL-phase separation is possible (and was always observed with O:C smaller 0.5 based on You et al. 2014) and depends on the organic used. As for phthalic acid ($C_8H_6O_4$), O:C is 0.5, how do you know that the particles actually are well mixed in section 2.1.1 after humidification? This is critical for the whole study and this assumption should be justified.

Regarding section 2.1.1. As solid crystalline ammonium sulfate is not spherical, how much this affects the estimation of the coating thickness? Further, could the difference between measured GFs and modelled ones be explained by this uncertainty in the coating thickness and further in the calculated organic mass

fraction? Even though the coating thickness is calculated to give equivalent mass fraction of PA compared to the well mixed case, I would assume that even small uncertainty in the HTDMA measurements and thus in the coating thickness affects the amount of calculated PA mass fraction. The mass fraction should be marked also to the figure 3.

There are numerous English language mistakes, especially in the last half of the paper. I have marked some of them for specific comments below. However, I suggest careful reading and re-writing, especially for the section 4, as there were some paragraphs that I could not understand at all. Related to section 4 (Summary and conclusions), I find that the main results mentioned in the abstract (e.g. at high RH GF decreases as thickness of PA shell increases) and conclusions derived from those are completely missing.

**Specific/minor comments**:

The acronym HTDMA is never explained, even though HTDMA instrument/setup is major part of the study.

Lines 54-56: terms homogeneously mixed and well mixed are both used later in the paper and figures, which is confusing (I understood they mean the same thing). I would select one of them and use it throughout.

Line 55: …aerosol particles may be divided…

Line 59: …of the earlier studies…

Lines 69-71: verb is missing, I assume you mean something like …few laboratory studies have investigated the influence…

Line 76: change difference to different?

Line 93: correct typo in hygroscopicity

Line 104: …aerosol particles have an average…

Section 2.1: Temperature to what the particles are exposed to is not mentioned, please add it. Ambient temperature was mentioned later at some point, but please be more specific as the experiments should be repeatable by others.

Line 166:  …aerosols were pre-humidified…

Line 168: I am not sure if I understand what you mean by "Finally the conditioning core-shell particle" here (referring to "the conditioning"). Maybe just the core-shell particle or remaining core-shell particle?

Line 174: Is the equation (1) now from the AIOMFAC model mentioned in lines 117-122? If it is, I would also mention that here as by first reading time I was wondering from where Eq. (1) comes from.

Line 211: …AS in the…particles dissolves…

Lines 231 and in some other places: when reporting numbers, I believe the correct way is to use "and" before the last number (i.e. 100, 150 and 200nm instead of 100, 150, 200nm)

Line 246: literatures -> literature

Line 247: …observed a strong higher water uptake… -> unclear, possibly "observed higher water uptake"

Line 248: correct typo: mass rations -> mass ratios

Lines 259-260: very long sentence, hard to follow. Could you reformulate/have two sentences instead?

Line 276: change "it assumes" into "it is assumed that"

Line 276: please remind the reader about the morphology effect also here with a short sentence in addition to referring section 3.2.

Line 289: correct "is" to "are" as you have plural

Line 294: AP most likely typo, please correct

Line 298-299: The sentence is unclear, maybe remove "the"?

Lines 301-304: very long sentence, so what was actually observed?

Line 309: repetition (observation was observed). Maybe change to …a contracting observation was made…

Line 332-333: I do not understand the sentence starting "they found that…", please elaborate/correct sentence structure

Lines 338-342: Again, very long and complicated sentence, hard to follow. Please simplify.

Line 352: change "estimation of" into "estimator for"

Lines 357-359: It seems there is main verb missing?

Lines 360-361: "According to filed studies…a variety of organic aerosol particles were characterised in the atmosphere…" This seems a little off from the context. Or do you mean that also in other studies various particle properties have been characterised? By first reading it seems you are writing about identifying particles, which is not related to your study in that sense.

Lines 366-367: …to depend on the difference of influence of kinetic limitations". Please reduce the use of "of" for clarity.

Line 369: change "significant" to "important"?

Line 370: change "combining" to "combined"

Line 371: change "organics coating" to "organic coatings"

Line 370-375: starting from "understanding…". Very long sentence, please break into smaller sections & elaborate, as now it is really hard to understand

Line 377: are the humidity cycles or particles depending on the ambient RH history? Make this clear in the text.

**References**

You, Y. et al, International Reviews in Physical Chemistry, 2014, Vol. 33, No. 1, 43–77, http://dx.doi.org/10.1080/0144235X.2014.890786

---

## Author Comment (AC1) · 4 Nov 2020

**Response to comments by anonymous referee #1:**

*Wang et al investigated the effect of mixing structure on the water uptake of mixtures of ammonium sulfate and phthalic acid particles by taking homogeneously internal mixture and core-shell structure into account. The aerosol water uptake is an important factor to determine many atmospheric processes, such as multiphase reactions and visibility reduction. The hygroscopicity of inorganic and organic mixtures is not well understood. The authors emphasized the importance of the particle morphology and mixing structure for inorganic-organic mixture water uptake. I would like to recommendit to publish to ACP after some minor revisions.*

**Response:** We are grateful to referee #1 for comments and suggestions to improve our manuscript. We have implemented changes based on these comments in the revised manuscript. We repeat the specific points raised by the reviewer in italic font, followed by our response. The page numbers and lines mentioned are referring to the Atmospheric Chemistry and Physics Discussions (ACPD) version.

**Comments:**

*1. Regarding the core-shell structure, the particle morphology may change when the particles are exposed to a high relative humidity condition. This could attribute to both core and shell take up enough water and get mixed together. Thus, the core-shell structure could be ambiguous.*

**Response:** Thanks. The reviewer is right, it is difficult to directly determine the morphology and the mixing state of aerosol particles at the high relative humidity (RH) when both core and shell components adsorb water. We tried to use several possible approaches for characterizing the morphology and mixing state of ammonium sulfate (AS) with phthalic acid (PA) in the RH range from 5 to 90 %: (1) utilize observational techniques such as optical microscopy and TEM; (2) assess the possibility of the presence of a non-well-mixed state utilizing a simplified model of the prevalence of liquid-liquid phase separation as a function of O:C ratio of the organic component and RH such as that described by You et al. (2013); (3) perform thermodynamic equilibrium calculations to model phase state as a function of water content (e.g., Song et al., 2012; Zuend and Seinfeld, 2013; Hodas et al., 2015). The detailed information has been described as follows:

(1) Although optical microscopy provides information on the morphology and the mixing state of particles, optical microscopy is applicable to micron-sized particles. Electron microscopy

technique (e.g., scanning electron microscopy (SEM) and transmission electron microscopy (TEM)) can characterize the morphology and the mixing state of sub-micron particles (Tang et al., 2019). However, the limitation of this technique is the vacuum condition, which is difficult to apply to measure the particle morphology and the mixing state at high RH. When collected core-shell particles at the high RH are exposed to the vacuum condition, it may affect their original physical state and mixing state (Tang et al., 2019).

(2) You et al. (2013) suggested that the liquid-liquid phase separation in particles containing organics and inorganic salts may occur when the O:C ratio of organic components is within 0.5 < O:C < 0.8 based on their observations. In our study, the ratio of O:C in the phthalic acid components in the AS/PA mixtures is ~ 0.5, which is possible to occur LLPS during humidification processes.

(3) Zuend group (Zuend and Seinfeld, 2013) has carried out the Aerosol Inorganic-Organic Mixtures Functional groups Activity Coefficients (AIOMFAC) gas-particle equilibrium calculations for core-shell AS/PA system. In the case of core-shell 200 nm AS coated by 50 nm PA aerosol particles, AIOMFAC model predicts that PA is assumed to be phase separated from AS all the time. Figure R1 shows that the AIOMFAC predictions are not in good agreement with the core-shell particles with 46 wt % PA when AS is completely deliquesced at RH above 80 %. This suggests that the morphology of core-shell aerosol particles may change due to both of AS and PA uptake water at RH above 80 %. Therefore, the morphology/mixing state in the humidification process, especially at high RH is still to be investigated.

[Figure]

Figure R1. Hygroscopic growth factor for core-shell for 200nm ammonium sulfate (AS) core with coating 50nm phthalic acid (PA) (black square). In comparison, the AIOMFAC prediction for growth factor of core-shell AS/PA aerosol particles with considering liquid-liquid phase separation (LLPS) (red line).

*2. The kinetic limitation has been referred for explaining the effect of core-shell structure on particle water uptake. In HTDMA system, the residence time of particles passing through the conditioned part may be short and lead to a kinetic limitation. This could not be the case when RH exceeds the DRH and particles are liquefied. We should also note that the particles with core-shell structure may have enough time to eliminate such kinetic limitation in the real atmosphere, different from the situation in the HTDMA system.*

**Response:** Thanks for the comment. The reviewer is right, the equilibrium time may be inadequate for equilibrium particle growth/shrink such as those coated with organic layers, which results in kinetic limitation. The transportation of water molecules into the bulk of particles and droplets can be kinetically limited by surface processes (e.g., surface accommodation and surface-bulk transfer) or by diffusion in the particle bulk (Seinfeld and Pandis, 2006; Taraniuk et al., 2007; Pöschl et al., 2007). For example, in case of the highly viscous substances, water absorption is likely to be restricted by bulk diffusion, leading to kinetically limited and gradual deliquescence transitions (Chan and Chan, 2005; Mikhailov et al., 2009). Also, organic coating can act as a physical barrier to prevent the water condensation/evaporation rate on droplet surfaces (Gill et al., 1983; Barnes, 1986) and can lower the accommodation coefficient (Pandis et al., 1995). Chuang (2003) suggested that these atmospheric particles exhibit longer equilibrium times for particles with a coating of an organic film with an approximate accommodation coefficient in the range of $1 \times 10^{-5}$ to $4 \times 10^{-5}$. Therefore, next, potential kinetic limitations in the HTDMA-measured hygroscopicity of core-shell aerosol particles is to be investigated in both humidification and dehumidification conditions.

The atmospheric aerosol particles are more likely internally mixed with organic fractions (e.g., well-mixed, core-shell mixing state), which are in equilibrium with surrounding conditions (e.g., temperature, RH, and reactive species (Bones et al., 2012; lei et al., 2014). Due to the complex physical chemistry of aerosol particles, e.g., amorphous aerosol particles refer to the rubber, gel, glassy, and viscous liquid, it is difficult and impossible to equilibrate water vapor in a short time (Chan and Chan, 2005: Mikhailov et al., 2009; Bones et al., 2012). For example, Bones et al. (2012) observed secondary organic compounds in the glassy state have mass transfer limitation in

equilibrium with surrounding water vapor. They further found that the timescale can be »$10^3$ for water equilibration with particles containing sucrose/sodium chloride/aqueous droplets as a proxy for multicomponent ambient aerosol. Also, due to complex atmospheric condition (i.e., wind, reaction, RH, T affect particle mixing state), their mixing state may change as aerosol particles are exposed to the real atmosphere (Riemer et al., 2019). Therefore, we need to comprehensively understand the kinetic limitations that may control water partitioning in ambient particles, especially for the amorphous aerosol particles.

*3. In addition, if the phthalic acid is surface active? The reduction in surface tension may be closely related to the hygroscopicity of core-shell particles.*

**Response:** Thanks for the comment. The phthalic acid is not surface active according to the previous reference (Padró et al., 2007). For the aerosol particles containing organic with surface active system, the hygroscopicity of aerosol particles may be enhanced by surface-active components. Ruehl et al. (2012) studied on droplet surface tension of mixed organic-inorganic particles at high RH (99.3–99.9 %). They found that there is 50-75 % reduction for the surface tension of droplets containing NaCl and a-pinene ozonolysis products, but only when enough organic material was present to coat on the droplet surface at least 0.8 nm thickness. They suggest that if surface-active particles account up more than 80 % in the atmosphere, their effect on cloud properties and thus climate could be enhanced (Padró et al., 2007):

*4. There are numerous English grammar errors, especially in the summary and conclusion section (Line 354-360).*

**Response:** Thanks for the comment. We have carefully revised the whole manuscript regarding language issues, including grammar, wording, and sentence structure, following the reviewer's suggestions, we have rewritten Sect. 4 (Summary and conclusion) they now reads as:

**Page 16 line 345-Page 18 line 381:** "In this study, we focused on PA to represent common organic compounds produced by various sources (e.g., vehicles, biomass burning, photo-oxidation). It is found that PA aerosol particles uptake water continuously as RH increases. We further investigated the effect of PA coating on the hygroscopicity of core-shell-generated aerosol particles. As PA coating thickness increases, the hygroscopic growth factor of AS/PA core-shell-generated particles increases prior to the deliquescence of AS, but the water uptake decreases at RH above 80 %. Furthermore, we compared the hygroscopic behavior of AS/PA core-shell-generated particles with

that of AS/PA initially well-mixed particles. Due to mixing state effects, higher hygroscopic growth factors of AS/PA core-shell-generated particles, compared to that of initially well-mixed particles, were observed in this study at RH above 80 %. In addition, the ZSR relation prediction is in good agreement with measured results of AS/PA initially well-mixed particles, but leads to the underestimation of the hygroscopic growth factor of AS/PA core-shell-generated particles at RH above 80 %. We attribute these discrepancies to the morphology effect when AS deliquesces in the core-shell-generated particles.

There are a vast number of internally mixed organic-inorganic aerosol particles existing in the atmosphere. The hygroscopicity behavior of mixture particles exhibits variability during RH cycles depending on the chemical composition, size, and mixing state. Humidity cycles may lead to liquid-liquid phase separation, e.g., in the form of core-shell aerosol particles, including at higher RH or in the salt-supersaturated concentration range. Also, due to the different physicochemical properties of organic compounds (e.g., viscosity, solubility, physical state, and morphology), the equilibrium time varies for these organic coated with inorganic aerosol particles. Therefore, potential kinetic limitations in the HTDMA-measured hygroscopicity of core-shell aerosol particles is to be investigated in both humidification and dehumidification conditions."

**Related changes included in the revised manuscript:**
In this study, we have systematically investigated the hygroscopicity of AS/PA aerosol particles with different mass fractions of PA in the different mixing states in terms of initial particle generation. Therefore, we revised "well mixed" as "initially well-mixed" in the response to referee files and the whole paper, including figures. Also, we added the explanation for the term "initially well-mixed aerosol particles". To further avoid the misleading terminality "core-shell", we revised "core-shell" as "core-shell-generated" in response to referee files and the whole paper, including figures.

**Page 3 line 58-61:** "Most of the previous studies on the hygroscopic behavior of multi-components aerosol focus on the well-mixed particles generated from initially well-mixed solutions (Miñambres et al., 2010; Shi et al., 2014; Gupta et al., 2015; Jing et al., 2016; Lei et al., 2014; 2018).

**Page 7 line 141:** "2.1.1 Initially well-mixed AS/PA Aerosol particles"

**Page 7 line 154:** "2.1.2 Core-shell-generated AS/PA aerosol particles"

**Page 9 line 188:** "3.1 Hygroscopic growth of initially well-mixed aerosol particles"

**Page 13 line 281-282:** "3.3 Comparison of core-shell-generated and initially well-mixed AS/PA aerosol particles"

**Page 14 line 291-293:** "At 75 % RH, the measured growth factor value of core-shell-generated particles is lower than that of initially well-mixed mixtures in the PA mass fraction range from 68 to 46 wt % due to the mass transfer limitations of water vapor transport to the AS core in the core-shell particles."

**Page 2 line 29-30:** "For the AS/PA initially well-mixed particles, a shift of deliquescence relative humidity (DRH) of AS (~80 %, Tang and Munkelwitz (1994)) to lower relative humidity (RH) is observed due to the presence of PA in the initially well-mixed particles."

**Page 3 line 61-64:** "For example, Choi and Chan (2002) studied on the effects of glycerol, succinic acid, malonic acid, citric acid, and glutaric acid on the hygroscopic properties of sodium chloride and AS in the initially well-mixed aerosol particles, respectively, using an electrodynamic balance."

**Page 14 line 288-290:** "However, compared to Fig 5a-b, Fig. 5c shows the hygroscopic growth factors of initially well-mixed AS/PA is slightly higher than that of AS/PA core-shell-generated particles with 46 wt % PA."

**Page 14 line 293-294:** "For the initially well-mixed AS/PA particles, however, partial dissolution of AS into the liquid AP phase may lead to more water uptake by initially well-mixed particles."

**Page 31 line 660-662:** "In comparison, the E-AIM model, the fitted expression Eq. (1), and the ZSR relation predicted growth factors of ammonium sulfate (AS), PA, and initially well-mixed particles with different mass fractions of PA, respectively."

**Page 34 line 694-697:** "In comparison, the E-AIM model, the fitted expression Eq. (1), and the ZSR relation predicted growth factors of ammonium sulfate (AS), PA, and initially well-mixed particles with different mass fractions of PA, respectively."

**Page 3 line 54-58:** We revised as "The initially well-mixed aerosol particles may be divided into homogeneous and heterogeneous internally mixed aerosol particles (Lang-Yona et al., 2009), which could, in turn, strongly influence the water uptake, optical properties, and the cloud condensation nuclei (CCN) ability of the particles (Lesins et al., 2002; Falkovich et al., 2004; Zhang et al., 2005; Schwarz et al., 2006; Su et al., 2010)."

**Page 4 line 69-72:** We revised as "However, to date, few laboratory studies have been investigated on the influence of organic coatings on the hygroscopic behavior of core-shell particles and the difference of mixing state effects on the hygroscopicity of aerosol particles (Zhang et al., 2008; Pagels et al., 2009; Xue et al., 2009; Lang-Yona et al., 2010; Ditas et al., 2018)."

**Page 4 line 76-77:** We revised as "They suggest that different organic coatings lead to changes in the hygroscopic properties of core-shell-generated particles."

**Page 5 line 92-94:** We revised as "The organic PA can have a profound effect on light scattering, hygroscopicity, and phase transition properties of multicomponent atmospheric aerosols."

**Page 8 line 161:** We added "The temperature required for vaporizing PA is between ~100 and ~130°C, which corresponds to coating thickness between 10 and 50 nm."

**Page 8 line 166-167:** We revised as "After core-shell-generated particle-sizing, aerosols were pre-humidified in a Nafion tube and flowed into the second Nafion humidifier at the set RH2 to reach equilibrium for growth of aerosol particles."

**Page 8 line 167-168:** We revised as "Finally, the humidified core-shell-generated particles were detected by a DMA3 and a CPC at ambient temperature."

**Page 10 line 209-212:** We revised as "For example, in the case of 1:3 mixtures of AS:PA (by mass), 75 wt % PA in the initially well-mixed particles suppresses the deliquescence of AS, i.e., AS in the initially well-mixed particles slowly dissolves into the liquid phase due to continuous water uptake of PA prior to DRH of AS (80 % RH)."

**Page 8 line 156-158:** We revised as "After a passage through a silica gel diffusion dryer and a neutralizer, the AS core aerosol particles with a certain diameter (100, 150, and 200 nm, respectively) were firstly selected by a DMA1 and then exposed to organic vapors in a coating system."

**Page 9 line 193-195:** We revised as "However, an abrupt increase in the hygroscopic growth factor is observed at 75 % RH for initially well-mixed particles containing 50 and 75 wt % PA, of which the growth factor is higher than that of pure PA aerosol particles (1.09 ± 0.01 nm from measurements shown in Fig. 2) at the same RH."

**Page 10 line 201-204:** We revised as "For example, the measured growth factors for initially well-mixed containing 25, 50, and 75 wt % PA are 1.36, 1.28, and 1.19 at 80 % RH, respectively, lower than the growth factor of 1.45 for pure deliquesced AS particles (value from measurements shown in Fig. 2) at the same RH."

**Page 11 line 230-231:** We revised as "Here, we investigated the hygroscopic behavior of samples of various AS core particle sizes (AS particle dry diameter of 100, 150, and 200 nm) and coating (PA coating of 10, 20, 30, and 50 nm), respectively."

**Page 11 line 237-239:** We revised as "For example, the measured growth factor value at 80 % RH is 1.45, 1.40, 1.32, and 1.28 for core-shell-generated particles containing 100 nm AS and 10, 20, 30, and 50 nm coating PA shell, respectively."

**Page 12 line 265-Page 13 line 268:** We revised as "In the case of 50-nm PA shell coated with a certain size of the AS core (100, 150, and 200 nm) with respect to 68, 55, and 46 wt % PA in the core-shell-generated particles, it exhibits an increase in hygroscopic growth factor of core-shell-generated particles at RH below 80 % as the size of AS core decreases."

**Page 12 line 246-247:** We revised as "The underprediction of the ZSR relation was also observed in the literature (Chan et al., 2006; Sjogren et al., 2007)."

**Page 12 line 246-247:** We revised as "Sjogren et al. (2006) observed an enhanced water uptake of mixtures consisting of AS and adipic acid with different mass ratios (1:2, 1:3, and 1:4) at RH above 80 % compared with ZSR relation in the hydration condition."

**Page 13 line 276-278:** We revised as "For ZSR prediction, it is assumed that volume fraction of AS components is constant according to the ratio of the volume of AS core in the sphere to the volume of a core-shell sphere based on Eq. (3)."

**Page 14 line 288-290:** We revised as "However, compared to Fig 5a-b, Fig. 5c shows the hygroscopic growth factors of initially well-mixed AS/PA are slightly higher than that of AS/PA core-shell-generated particles with 46 wt % PA."

**Page 14 line 293-294:** We revised as "For the initially well mixed AS/PA particles, however, partial dissolution of AS into the liquid PA phase may lead to more water uptake by initially well-mixed particles."

**Page 14 line 293-294:** We revised "Core-shell-generated particle morphology may experience the restructuring and associate size change of particles." as

"Accordingly, at high RH, the occurrence of microscopic restructuring of core-shell-generated particles may affect their size."

**Page 14 line 301-304:** We revised as "Chan et al. (2006) investigated hygroscopicity of 49 wt % glutaric acid coated on AS core during two continuous hydration cycles. They observed that the experimental growth factor of the fresh core-shell of AS and glutaric acid in the first hydration cycle is slightly higher than those in the second hydration cycle with the same mass fractions of glutaric acid."

**Page 14 line 309-Page 15 line 310:** We revised as "However, a contrasting phenomenon was observed in the previous study (Maskey et al., 2014)."

**Page 14 line 332-334:** We revised as "they found that the slightly higher growth factors of initially well-mixed particles are than that of core-shell-generated aerosol particles (12-nm levoglucosan coated 88-nm AS)."

**Page 16 line 338-342:** We revised as "In addition, for the AS/PA mixture aerosol particles containing 46-68 wt % PA, the measured growth factors of initially well-mixed AS/PA particles are in good agreement with the ZSR relation prediction comparing with that of core-shell-generated AS/PA particles."

**Reference:**

Barnes, G. T.: The effects of monolayers on the evaporation of liquids, Adv. Colloid Interface Sci., 25, 89–200, 1986.

Bones, D. L., Reid, J. P., Lienhard, D. M., and Krieger, U. K.: Comparing the mechanism of water condensation and evaporation in glassy aerosol, Proceedings of the National Academy of Sciences of the United States of America, 109, 11613-11618, 2012.

Chan, M. N. C. a. C. K.: Mass transfer effects in hygroscopic measurements of aerosol particles, Atmospheric Chemistry and Physics, 2005. 2005.

Chuang, P.Y.: Measurement of the timescale of hygroscopic growth for atmospheric aerosols, J. Geophys. Res., 108(D9), 4282, doi:10.1029/2002JD002757, 2003.

Gill, R. S., Graedel, T. E., and Weschler, C. J.: Organic films on atmospheric aerosol particles, fog droplets, cloud droplets, raindrops, and snowflakes, Rev. Geophys., 22, 903–920, 1983.

Lei, T., Zuend, A., Wang, W. G., Zhang, Y. H., and Ge, M. F.: Hygroscopicity of organic compounds from biomass burning and their influence on the water uptake of mixed organic ammonium sulfate aerosols, Atmos. Chem. Phys., 14, 11165-11183, 2014.

Mikhailov, E., Vlasenko, S., Martin, S. T., Koop, T., and Poschl, U.: Amorphous and crystalline aerosol particles interacting with water vapor: conceptual framework and experimental evidence for restructuring, phase transitions and kinetic limitations, Atmospheric Chemistry and Physics, 9, 9491-9522, 2009.

Padró, L. T., Asa-Awuku, A., Morrison, R., and Nenes, A.: Inferring thermodynamic properties from CCN activation experiments: single-component and binary aerosols, Atmos. Chem. Phys., 7, 5263-5274, 2007.

Pandis, S. N., Wexler, A. S., and Seinfeld, J. H.: Dynamics of tropospheric aerosols, J. Phys. Chem., 99, 9646–9659, 1995.

Pöschl, U., Rudich,Y., and Ammann, M.: Kinetic model framework for aerosol and cloud surface chemistry and gas-particle interactions – Part 1: General equations, parameters, and terminology, Atmos. Chem. Phys., 7, 5989–6023, 2007, http://www.atmos-chem-phys.net/7/5989/2007/.

Seinfeld, J. H. and Pandis, S. N.: Atmospheric chemistry and physics, J.Wiley & Sons, New York, 2006.

Riemer, N., Ault, A. P., West, M., Craig, R. L., and Curtis, J. H.: Aerosol Mixing State: Measurements, Modeling, and Impacts, Reviews of Geophysics, 57, 187-249, 2019.

Song, M., Marcolli, C., Krieger, U. K., Zuend, A., and Peter, T.: Liquid-liquid phase separation and morphology of internally mixed dicarboxylic acids/ammonium sulfate/water particles, Atmos. Chem. Phys., 12, 2691-2712, https://doi.org/10.5194/acp-12-2691-2012, 2012.

Taraniuk, I., Graber, E. R., Kostinski, A., and Rudich,Y.: Surfactant properties of atmospheric and model humic-like substances (HULIS), Geophys. Res. Lett., 34, L16807, doi:10.1029/2007GLo29576, 2007.

You, Y., Renbaum-Wolff, L., and Bertram, A. K.: Liquid–liquid phase separation in particles containing organics mixed with ammonium sulfate, ammonium bisulfate, ammonium nitrate or sodium chloride, Atmos. Chem. Phys., 13, 11723-11734, https://doi.org/10.5194/acp-13-11723-2013, 2013.

Zuend, A. and Seinfeld, J. H.: A practical method for the calculation of liquid-liquid equilibria in multicomponent organic-water-electrolyte systems using physiochemical constraints, Fluid Phase Equilibr., 337, 201–213, 2013.

---

## Author Comment (AC2) · 4 Nov 2020

**Response to comments by anonymous referee #2:**

*Wang and Lei et al., present a study about the hygroscopicity of aerosol particles consisting of ammonium sulfate (AS) and phthalic acid (PA). Using a HTDMA setup, the authors study first the hygroscopicity of particles where AS and PA are present in a well-mixed, internal mixture and then particles consisting of AS core and PA shell of varying size. Later, the authors compare the hygroscopicity of internally mixed particles to those with core-shell structure.*

*The authors show that at RH above 80% the core-shell particles have higher hygrocopicity than the well-mixed particles. Further, a traditional ZSR-relation, coupled with an empirical growth factor fit for PA hydration curve, predicts a lower hygroscopicity than what is measured for the core-shell particles. These differences in predicted and measured hygroscopicity are attributed to particle morphology changes, i.e., the shape of the particles deviate from a spherical shape.*

*The manuscript is generally well written and the results increase the understanding of Atmospheric Chemistry and Physics community about the hygroscopicity of complex organic-inorganic particles. I recommend that the manuscript is published after a minor revision.*

**Response:** We are grateful to referee #2 for comments and suggestions to improve our manuscript. We have implemented changes based on these comments in the revised manuscript. We repeat the specific points raised by the reviewer in italic font, followed by our response. The page numbers and lines mentioned are with respect to the Atmospheric Chemistry and Physics Discussions (ACPD) version.

**General comments:**

*1. How the uncertainty of the measurements was determined? In Figures 2-5 the measured points have uncertainty both in the RH and growth factor direction. However, in the manuscript no information is given how this uncertainty was calculated.*

**Response:** Thanks for your comment. The measured results have the uncertainty of both RH and the hygroscopic growth factor as shown in Figures 2-5. The RH uncertainty is the accuracy of RH sensor ($\pm 2$ %). We calculated the uncertainty of growth factor of aerosol particles in the RH range from 5 to 90 % according to the following formula (Mochida and Kawamura, 2004):

$$\sqrt{\left(\left(g_f \frac{\sqrt{2}\varepsilon_{Dp}}{D_p}\right)^2 + \left(\varepsilon_{RH} \frac{dg_f}{dRH}\right)^2,\right)} \qquad\qquad (R1)$$

Here, $\varepsilon_{Dp}$, $\varepsilon_{RH}$, and $g_f$ are uncertainty of particle mobility diameter, relative humidity, and growth factor at different RHs, respectively. The average sizing offsets of our system is taken here as $\frac{\varepsilon_{Dp}}{D_p}$. Therefore, in our study, the calculated uncertainty of growth factor of AS/PA aerosol particles is ~1 %-2 %.

**Related additions included in the revised manuscript:**

**Page 7 line 137: we add** "Thus, the calculated uncertainty of growth factor depends on the error

propagation formula by $\sqrt{\left(\left(g_f \frac{\sqrt{2}\varepsilon_{Dp}}{D_p}\right)^2 + \left(\varepsilon_{RH} \frac{dg_f}{dRH}\right)^2,\right)}$ (Mochida and Kawamura, 2004). Here,

$\varepsilon_{Dp}$, $\varepsilon_{RH}$, and $g_f$ are uncertainty of particle mobility diameter, relative humidity, and growth factor at different RH, respectively. The average sizing offsets of our system is taken here as $\frac{\varepsilon_{Dp}}{D_p}$. Also, the RH uncertainty is the accuracy of RH sensor ($\pm 2$ %). In this study, the calculated uncertainty of growth factor of AS/PA aerosol particles is ~1%-2%."

*Specific comments:*

*1. Section 2.2.2. Please explain what the different R symbols are in Eq. (2) and (3). Supposedly they refer to radii of spheres.*

**Response:** Thanks for the comment. "$R_{AS}$" is the radius of core. "$R_{core\text{-}shell}$" is the radius of core-shell aerosol.

**Related additions included in the revised manuscript:**

**Page 9 line 181: We added** "$R_{As}$ is the radius of core and $R_{core\text{-}shell}$ is the radius of core-shell-generated aerosol."

*2. Lines 242–244. To me it looks like the ZSR relation does not predict the hygroscopic growth of AS/PA core-shell particles. The sentence starting from line 245 states the same ("The ZSR-based predictions are lower than...")*

**Response:** Thanks for comment. The predicted growth factor of core-shell-generated AS/PA aerosol particles by the ZSR relation is lower than the measured results, suggesting an unfavorable

assumption of ZSR relation (i.e., additivity of water uptake by the different mixture components according to their individual hygroscopicity). The range of measurement–model comparisons presented in this study indicates that providing accurate thermodynamic model predictions of the hygroscopic growth behavior of core-shell systems remains a challenging problem.

*3. Lines 338–342. I do not understand this sentence. Please rephrase it. Do you mean that in the future you will explore why the ZSR relation predicts lower growth factors than what is measured for the core-shell particles?*

**Response:** Thanks for comment. No, "The low mass fraction of PA will be explored in future" means: "In the sect. 3.3, we have measured and compared the hygroscopic growth factor of core-shell-generated AS/PA particles with that of initially well-mixed aerosol particles with the same mass fraction of PA (46-68 wt %). Next, the hygroscopic growth factor of core-shell AS/PA particles containing less than 46 wt % PA (e.g., 20, or 10 wt % PA in the core-shell) would be investigated and compared to that of well-mixed particles, and ZSR relation prediction."

**Related additions and changes included in the revised manuscript:**

**Page 16 line 338-342: we revised sentences as** "In addition, for the AS/PA mixture aerosol particles containing 46-68 wt % PA, the measured growth factors of well-mixed AS/PA particles are in good agreement with the ZSR relation prediction comparing with that of core-shell-generated AS/PA particles."

*4. The manuscript contains several typographical or grammar errors.*

**Response:** Thanks for the comment. We have carefully revised the whole manuscript regarding language issues, including grammar, wording, and sentence structure, following the reviewer's suggestions, we have rewritten Sect. 4 (Summary and conclusion) they now reads as:

**Page 16 line 345-Page 18 line 381:** "In this study, we focused on PA to represent common organic compounds produced by various sources (e.g., vehicles, biomass burning, photo-oxidation). It is found that PA aerosol particles uptake water continuously as RH increases. We further investigated the effect of PA coating on the hygroscopicity of core-shell-generated aerosol particles. As PA coating thickness increases, the hygroscopic growth factor of AS/PA core-shell-generated particles increases prior to the deliquescence of AS, but the water uptake decreases at RH above 80 %. Furthermore, we compared the hygroscopic behavior of AS/PA core-shell-generated particles with that of AS/PA initially well-mixed particles. Due to mixing state effects, higher hygroscopic

growth factors of AS/PA core-shell-generated particles, compared to that of initially well-mixed particles, were observed in this study at RH above 80 %. In addition, the ZSR relation prediction is in good agreement with measured results of AS/PA initially well-mixed particles, but leads to the underestimation of the hygroscopic growth factor of AS/PA core-shell-generated particles at RH above 80 %. We attribute these discrepancies to the morphology effect when AS deliquesces in the core-shell-generated particles.

There are a vast number of internally mixed organic-inorganic aerosol particles existing in the atmosphere. The hygroscopicity behavior of mixture particles exhibits variability during RH cycles depending on the chemical composition, size, and mixing state. Humidity cycles may lead to liquid-liquid phase separation, e.g., in the form of core-shell aerosol particles, including at higher RH or in the salt-supersaturated concentration range. Also, due to the different physicochemical properties of organic compounds (e.g., viscosity, solubility, physical state, and morphology), the equilibrium time varies for these organic coated with inorganic aerosol particles. Therefore, potential kinetic limitations in the HTDMA-measured hygroscopicity of core-shell aerosol particles is to be investigated in both humidification and dehumidification conditions."

**Related changes included in the revised manuscript:**

In this study, we have systematically investigated the hygroscopicity of AS/PA aerosol particles with different mass fractions of PA in the different mixing states in terms of initial particle generation. Therefore, we revised "well mixed" as "initially well-mixed" in the response to referee files and the whole paper, including figures. Also, we added the explanation for the term "initially well-mixed aerosol particles". To further avoid the misleading terminality "core-shell", we revised "core-shell" as "core-shell-generated" in response to referee files and the whole paper, including figures.

**Page 3 line 58-61:** "Most of the previous studies on the hygroscopic behavior of multi-components aerosol focus on the well-mixed particles generated from initially well-mixed solutions (Miñambres et al., 2010; Shi et al., 2014; Gupta et al., 2015; Jing et al., 2016; Lei et al., 2014; 2018).

**Page 7 line 141:** "2.1.1 Initially well-mixed AS/PA Aerosol particles"

**Page 7 line 154:** "2.1.2 Core-shell-generated AS/PA aerosol particles"

**Page 9 line 188:** "3.1 Hygroscopic growth of initially well-mixed aerosol particles"

**Page 13 line 281-282:** "3.3 Comparison of core-shell-generated and initially well-mixed AS/PA aerosol particles"

**Page 14 line 291-293:** "At 75 % RH, the measured growth factor value of core-shell-generated particles is lower than that of initially well-mixed mixtures in the PA mass fraction range from 68 to 46 wt % due to the mass transfer limitations of water vapor transport to the AS core in the core-shell particles."

**Page 2 line 29-30:** "For the AS/PA initially well-mixed particles, a shift of deliquescence relative humidity (DRH) of AS (~80 %, Tang and Munkelwitz (1994)) to lower relative humidity (RH) is observed due to the presence of PA in the initially well-mixed particles."

**Page 3 line 61-64:** "For example, Choi and Chan (2002) studied on the effects of glycerol, succinic acid, malonic acid, citric acid, and glutaric acid on the hygroscopic properties of sodium chloride and AS in the initially well-mixed aerosol particles, respectively, using an electrodynamic balance."

**Page 14 line 288-290:** "However, compared to Fig 5a-b, Fig. 5c shows the hygroscopic growth factors of initially well-mixed AS/PA is slightly higher than that of AS/PA core-shell-generated particles with 46 wt % PA."

**Page 14 line 293-294:** "For the initially well-mixed AS/PA particles, however, partial dissolution of AS into the liquid AP phase may lead to more water uptake by initially well-mixed particles."

**Page 31 line 660-662:** "In comparison, the E-AIM model, the fitted expression Eq. (1), and the ZSR relation predicted growth factors of ammonium sulfate (AS), PA, and initially well-mixed particles with different mass fractions of PA, respectively."

**Page 34 line 694-697:** "In comparison, the E-AIM model, the fitted expression Eq. (1), and the ZSR relation predicted growth factors of ammonium sulfate (AS), PA, and initially well-mixed particles with different mass fractions of PA, respectively."

**Page 3 line 54-58:** We revised as "The initially well-mixed aerosol particles may be divided into homogeneous and heterogeneous internally mixed aerosol particles (Lang-Yona et al., 2009), which could, in turn, strongly influence the water uptake, optical properties, and the cloud condensation nuclei (CCN) ability of the particles (Lesins et al., 2002; Falkovich et al., 2004; Zhang et al., 2005; Schwarz et al., 2006; Su et al., 2010)."

**Page 4 line 69-72:** We revised as "However, to date, few laboratory studies have been investigated on the influence of organic coatings on the hygroscopic behavior of core-shell particles and the difference of mixing state effects on the hygroscopicity of aerosol particles (Zhang et al., 2008; Pagels et al., 2009; Xue et al., 2009; Lang-Yona et al., 2010; Ditas et al., 2018)."

**Page 4 line 76-77:** We revised as "They suggest that different organic coatings lead to changes in the hygroscopic properties of core-shell-generated particles."

**Page 5 line 92-94:** We revised as "The organic PA can have a profound effect on light scattering, hygroscopicity, and phase transition properties of multicomponent atmospheric aerosols."

**Page 8 line 161:** We added "The temperature required for vaporizing PA is between ~100 and ~130°C, which corresponds to coating thickness between 10 and 50 nm."

**Page 8 line 166-167:** We revised as "After core-shell-generated particle-sizing, aerosols were pre-humidified in a Nafion tube and flowed into the second Nafion humidifier at the set RH2 to reach equilibrium for growth of aerosol particles."

**Page 8 line 167-168:** We revised as "Finally, the humidified core-shell-generated particles were detected by a DMA3 and a CPC at ambient temperature."

**Page 10 line 209-212:** We revised as "For example, in the case of 1:3 mixtures of AS:PA (by mass), 75 wt % PA in the initially well-mixed particles suppresses the deliquescence of AS, i.e., AS in the initially well-mixed particles slowly dissolves into the liquid phase due to continuous water uptake of PA prior to DRH of AS (80 % RH)."

**Page 8 line 156-158:** We revised as "After a passage through a silica gel diffusion dryer and a neutralizer, the AS core aerosol particles with a certain diameter (100, 150, and 200 nm, respectively) were firstly selected by a DMA1 and then exposed to organic vapors in a coating system."

**Page 9 line 193-195:** We revised as "However, an abrupt increase in the hygroscopic growth factor is observed at 75 % RH for initially well-mixed particles containing 50 and 75 wt % PA, of which the growth factor is higher than that of pure PA aerosol particles (1.09 ± 0.01 nm from measurements shown in Fig. 2) at the same RH."

**Page 10 line 201-204:** We revised as "For example, the measured growth factors for initially well-mixed containing 25, 50, and 75 wt % PA are 1.36, 1.28, and 1.19 at 80 % RH, respectively, lower than the growth factor of 1.45 for pure deliquesced AS particles (value from measurements shown in Fig. 2) at the same RH."

**Page 11 line 230-231:** We revised as "Here, we investigated the hygroscopic behavior of samples of various AS core particle sizes (AS particle dry diameter of 100, 150, and 200 nm) and coating (PA coating of 10, 20, 30, and 50 nm), respectively."

**Page 11 line 237-239:** We revised as "For example, the measured growth factor value at 80 % RH is 1.45, 1.40, 1.32, and 1.28 for core-shell-generated particles containing 100 nm AS and 10, 20, 30, and 50 nm coating PA shell, respectively."

**Page 12 line 265-Page 13 line 268:** We revised as "In the case of 50-nm PA shell coated with a certain size of the AS core (100, 150, and 200 nm) with respect to 68, 55, and 46 wt % PA in the core-shell-generated particles, it exhibits an increase in hygroscopic growth factor of core-shell-generated particles at RH below 80 % as the size of AS core decreases."

**Page 12 line 246-247:** We revised as "The underprediction of the ZSR relation was also observed in the literature (Chan et al., 2006; Sjogren et al., 2007)."

**Page 12 line 246-247:** We revised as "Sjogren et al. (2006) observed an enhanced water uptake of mixtures consisting of AS and adipic acid with different mass ratios (1:2, 1:3, and 1:4) at RH above 80 % compared with ZSR relation in the hydration condition."

**Page 13 line 276-278:** We revised as "For ZSR prediction, it is assumed that volume fraction of AS components is constant according to the ratio of the volume of AS core in the sphere to the volume of a core-shell sphere based on Eq. (3)."

**Page 14 line 288-290:** We revised as "However, compared to Fig 5a-b, Fig. 5c shows the hygroscopic growth factors of initially well-mixed AS/PA are slightly higher than that of AS/PA core-shell-generated particles with 46 wt % PA."

**Page 14 line 293-294:** We revised as "For the initially well mixed AS/PA particles, however, partial dissolution of AS into the liquid PA phase may lead to more water uptake by initially well-mixed particles."

**Page 14 line 293-294:** We revised "Core-shell-generated particle morphology may experience the restructuring and associate size change of particles." as

"Accordingly, at high RH, the occurrence of microscopic restructuring of core-shell-generated particles may affect their size."

**Page 14 line 301-304:** We revised as "Chan et al. (2006) investigated hygroscopicity of 49 wt % glutaric acid coated on AS core during two continuous hydration cycles. They observed that the experimental growth factor of the fresh core-shell of AS and glutaric acid in the first hydration cycle is slightly higher than those in the second hydration cycle with the same mass fractions of glutaric acid."

**Page 14 line 309-Page 15 line 310:** We revised as "However, a contrasting phenomenon was observed in the previous study (Maskey et al., 2014)."

**Page 14 line 332-334:** We revised as "they found that the slightly higher growth factors of initially well-mixed particles are than that of core-shell-generated aerosol particles (12-nm levoglucosan coated 88-nm AS)."

**Page 16 line 338-342:** We revised as "In addition, for the AS/PA mixture aerosol particles containing 46-68 wt % PA, the measured growth factors of initially well-mixed AS/PA particles are in good agreement with the ZSR relation prediction comparing with that of core-shell-generated AS/PA particles."

**Reference:**

Mochida, M. and Kawamura, K.: Hygroscopic properties of levoglucosan and related organic compounds characteristic to biomass burning aerosol particles, Journal of Geophysical Research-Atmospheres, 109, 2004.

---

## Author Comment (AC3) · 4 Nov 2020

**Response to comments by anonymous referee #3:**

*Wang et al. investigated the effect of different internal mixing structures (homogeneously mixed and core-shell-generated structure) on the water uptake of aerosols consisting of ammonium sulfate (AS) and phthalic acid (PA). In addition, they studied how the amount of PA in the particle affect the water uptake. They used specific HTDMA-instrument to select specific sized particles, add PA coating into them (when core-shell-generated structures were studied) and humidify then following by measurement of the growth of the particles as they uptake water. To accompany the measured data, they used theory for estimating the hygroscopic growth of individual components and Zdanovskii-Stokes-Robinson (ZRS) relation to calculate the hygroscopic growth of mixed particles. For homogeneously mixed particles (also referred as well-mixed particles) they observed, for example, that a decrease in the hygroscopic growth factor (GF) with increasing mass fractions of PA at above RH 80% level. They state that these results also agreed with previous studies, and the predictions from ZRS also agreed rather well for the well-mixed particles. For the core-shell-generated structured particles, Wang et al. observed an increase in the GF as the size of the AS core decreased (below 80% RH level). At above 80% RH level, they observed a decrease in the GF with decreasing size of the AS core. For the core-shell-generated structured particles ZRS predictions underestimated the hygroscopic growth. As a general comment, the methods and experimental procedures are adequately described, and they seem valid for this type of study. My main criticism concentrates on to the relevance of the study, and what new information it brings to the scientific community.*

**Response:** We are grateful to referee #3 for comments and suggestions to improve our manuscript. We have implemented changes based on these comments in the revised manuscript. We repeat the specific points raised by the reviewer in italic font, followed by our response. The page numbers and lines mentioned are with respect to the Atmospheric Chemistry and Physics Discussions (ACPD) version.

**General comments:**

*1. Why liquid well mixed AS-PA (ammonium sulfate: AS, phthalic acid: PA) would have different hygroscopicity compared to the AS particle with PA coating? Or was this the research question of the study?*

**Response:** Many thanks. We observed a difference in the hygroscopic growth factor between AS/PA initially well-mixed particles and AS/PA core-shell particles generated by different methods. A similar phenomenon has been observed in previous studies (Chan et al., 2006; Maskey et al., 2014). We attribute the observed differences to mixing state effects. Therefore, the effect of mixing state on the hygroscopic behavior of AS/PA mixture particles has been investigated in our research.

*2. The introduction of the draft is strongly focused on the water uptake of the aerosols which is the main theme of the paper. The atmospheric relevance of phthalic acid (PA) is discussed in the introduction (lines 86-98) into some extent, and shortly mentioned in the conclusions (lines 347-349). However, as published studies about the hygroscopicity of organic coatings with inorganic core do exists could you provide more explanation what new this study brings and how PA it its relevant to the atmosphere in larger scale and thereby justify its use in this specific study? Can the results acquired for PA be generalised for other organic compounds too? For example, the authors mention that PA is common in rural mountains and marine atmospheres in Asia, so is it specific to those areas only? Could you, for example, give an estimate how much of PA there is in the atmosphere compared to other organic compounds? In addition, even though the use of ammonium sulfate (AS) is very common for this type of studies, its relevance/why it was used could also be shortly mentioned.*

**Response:** Many thanks. Many studies on water uptake of particles showing an organic coating with inorganic core were reported in the literature (Chan et al., 2006; Ciobanu et al., 2009; Song et al., 2012, 2018; Shiraiwa et al., 2013; Maskey et al., 2014; Hodas et al., 2015). For example, Song et al. (2012) observed a liquid-liquid phase separation of internally mixed dicarboxylic acids containing 5, 6, and 7 carbon atoms with ammonium sulfate during the humidification and dehumidification measurements. However, to date, there are only very few studies that have attempted to investigate an effect of coating thickness on the hygroscopicity of the core-shell particles, and further effects of mixing state on the hygroscopicity of aerosol particles (Maskey et al., 2014). In this study, we have systematically investigated the hygroscopicity of AS/PA aerosol particles with different mass fractions of PA in the different mixing states in terms of initial particle generation.

According to Wang et al. (2011), sampling of aerosol particles, including the water-soluble organic carbon (WSOC) fraction, was conducted in their field study. Among many organic acids, PA was detected in fine mode aerosols from the urban and remote mountain atmosphere of China and from the marine atmosphere in the outflow region of East Asia during field campaigns (Wang et al., 2011). They observed that PA is one of the most abundant aerosol components at the marine site during the measurements. Furthermore, these secondarily produced aromatic acids, which are photo-oxidation products from anthropogenic precursors such as toluene, xylene, and naphthalene, are initially formed as gaseous products and subsequently condensed onto pre-existing particles. Also, Kleindienst et al. (1999) investigated that hygroscopic particles formed after irradiating toluene, p-xylene, and 1,3,5-trimethylbenzene in the presence of $NO_x$ and AS seed particles in a chamber. This suggests the likely existence of PA in atmospheric aerosol particles. Organic coatings on inorganic aerosol particles in the atmosphere can play an important role in the range of RHs over which particle-bound water influences aerosol properties, such as density, light scattering, or refractive index. The field measurements showed that there is a substantial decrease in RH dependence of light scattering with increasing organic mass fraction (Varutbangkul et al., 2006). Therefore, an HTDMA study on effects of PA coating on the hygroscopicity of AS/PA core-shell aerosol particles was selected to investigate and simulate core-shell aerosol particles containing PA in the atmosphere.

Regarding a generalization of PA-specific effects to other organics, for the case of hygroscopic growth behavior of PA/AS core-shell-generated particles, we compared them with other organic/inorganic core-shell particles and we observed that there is no simple generalization to other organics coating an AS core. For example, in our study, we observed that the growth factors of AS/PA core-shell-generated particles are slightly higher than that of AS/PA initially well-mixed aerosol particles, which is contrasting with observations by Maskey et al. (2014). They found that the hygroscopic growth factors of AS/succinic acid and AS/levoglucosan well-mixed particles are higher than that of AS/succinic acid and AS/levoglucosan core-shell particles. The possible reasons for the difference between our study and results from Maskey et al. (2014) are the physical properties of the organic components, such as hygroscopicity, viscosity, volatility, and water solubility shown in Table R1.

Table R1. Solubility in pure water and bulk deliquescence relative humidity (DRH) of organic compounds at 25°C.

| component | Solubility (mol/kg) | Bulk DRH, % |
|---|---|---|
| Succinic acid | 0.49 | 98.8[a] |
| Phthalic acid | 0.04 | - |
| levoglucosan | 8.32 | 80[c] |

[a]Peng et al. (2001)

[b]Mochida and Kawamura (2004)

-: no report

For succinic acid, a moderately water-soluble dicarboxylic acid found in the atmosphere, the hygroscopicity of succinic acid aerosol particles has been investigated by many groups (Peng et al., 2001; Hämeri et al., 2002; Wise et al., 2003; Wex et al., 2007; Henning et al., 2012; Jing et al., 2016). They found no hygroscopic growth of initially solid, dry succinic acid aerosol particles as RH increases up to 95 %. Peng et al. (2001) measured the deliquescence relative humidity (DRH) of succinic acid using a bulk solution at 24°C, and its DRH is ~99 %. Also, Henning et al. (2012) observed no hygroscopicity of soot/succinic acid core-shell particles in the hydration mode using an HTDMA.

[Figure]

**Figure R1**. Hygroscopic diameter growth factor for 100 nm (dry diameter, RH < 5 %) succinic acid aerosol particles during humidification mode from 5 % RH to 90 % RH at 298 K (Jing et al., 2016).

Although the water-solubility of PA aerosol particles is lower compared with that of succinic acid, initially dry PA particles uptake water gradually in the whole RH range from 5 to 90 % (Brooks et al., 2004; Hämeri et al., 2002; Jing et al., 2016). Hämeri et al. (2002). Also, they investigated the

measured hygroscopic behavior of 100 nm aerosol particles consisting of ammonium sulfate and phthalic acid. Due to a 50 wt % PA components, a slight smoothing of the AS deliquescence behavior was observed in the hydration mode.

[Figure]

**Figure R2**. Hygroscopic diameter growth factor for 100 nm (dry diameter, RH < 5 %) phthalic acid aerosol particles during humidification mode from 5 % RH to 90 % RH at 298 K from this study.

Levoglucosan aerosol particles are of higher solubility than succinic acid. There is a gradual increase in growth factor as RH increases up to 98 %. No DRH of levoglucosan was observed in the hydration mode (Mochida and Kawamura 2004; Mikhailov et al., 2008; Lei et al., 2014, 2018).

[Figure]

**Figure R3**. Hygroscopic diameter growth factor for 100 nm (dry diameter, RH < 5 %) levoglucosan aerosol particles during humidification mode from 5 % RH to 90 % RH at 298 K (Lei et al., 2014).

In the case of AS/succinic acid mixed particles measured by Maskey et al. (2014), they found a higher growth factor for the well-mixed particles compared to their core-shell aerosol particles, which is contrary to our observations for AS/PA particles. A possible reason is the difference in the pure-component hygroscopic behavior of succinic acid vs. PA. No water uptake by AS/succinic acid core-shell particles was observed at RH lower than 80 % RH, while there is a gradual increase in water absorption of AS/PA core-shell-generated particles prior to the complete deliquescence of ammonium sulfate. This suggests that the physical state of the organic acid shell is solid for the succinic acid case by Maskey et al. (2014), while it is liquid-like in our study with PA. At RH above 80 %, a potential kinetic limitation on the water uptake through a sufficiently thick solid shell into the hygroscopic AS core is more obvious than one through a liquid organic shell into the core (i.e., liquid diffusion coefficient of water is the range of $10^{-10}$ to $10^{-9}$ $m^2$ $s^{-1}$, solid diffusion coefficient of water is the range of $10^{-13}$-$10^{-14}$ $m^2$ $s^{-1}$ at 25°C). This indicates that a sufficient residence time under humidification is required to reach gas-particle equilibrium of water for AS coated succinic acid.

In the case of AS/levoglucosan particles measured by Maskey et al. (2014), they found a slightly higher growth factor for the well-mixed particles compared to core-shell aerosol particles (80:20 by volume). In our study, AS/PA core-shell-generated particles containing the mass fraction range from 46 to 68 wt % have been investigated, which is different from the mass fraction of levoglucosan in the core-shell particles from Maskey et al. (2014). Therefore, different mass fraction of PA in the core-shell-generated particles may play a role in affecting the hygroscopicity of core-shell-generated particles.

Also, PA is common in rural mountains and marine atmospheres in Asia and is specific to these areas in the East Asian. According to Wang et al. (2011), sampling of aerosol particles, including the water-soluble organic carbon (WSOC) fraction, was conducted on 11–14 January, 12–20 February, and 12–24 April 2008 during their field study. There is no specific information on the mass fraction of PA in the atmospheric particle samples. According to Wang et al. (2011), PA is among the highest in mass concentration of individual secondary organic compounds detected in the urban and mountaintop air during winter and spring (Wang et al., 2011).

AS, a major component of the atmospheric aerosol, is chosen as a test substance because its thermodynamic behavior is well characterized. Most importantly, ammonium sulfate particles are stable and not volatile, while is important for reliable HTDMA measurements.

**Related changes included in the revised manuscript:**

**Page 5 line 91:** We added more description of the atmospheric relevance of PA there: "Furthermore, these aromatic acids like PA, which are photo-oxidation products of anthropogenic precursors such as toluene, xylene, and naphthalene, are initially formed as gaseous products and subsequently condensed onto pre-existing particles. Also, Kleindienst et al. (1999) investigated that hygroscopic particles formed after irradiating toluene, p-xylene, and 1,3,5-trimethylbenzene in the presence of NOx and AS seed in a chamber. This suggests the likely existence of PA in atmospheric aerosol particles. Organic coatings on inorganic aerosol particles in the atmosphere can play an important role in the range of RHs over which particle-bound water influences aerosol properties, such as the overall density, the light scattering behavior and the refractive index."

**Page 6 line 112:** We added more description of the atmospheric relevance of AS there: "AS is chosen as a test substance because it is a major component of the atmospheric aerosol and its thermodynamic behavior is well characterized. Most importantly, ammonium sulfate particles are stable and not volatile, useful features for HTDMA studies."

*3. As you note in the introduction (lines ), with 0.5 < O:C < 0.8 the LL-phase separation is possible (and was always observed with O:C smaller 0.5 based on You et al. 2014) and depends on the organic used. As for phthalic acid (C8H6O4), O:C is 0.5, how do you know that the particles actually are well mixed in section 2.1.1 after humidification? This is critical for the whole study and this assumption should be justified.*

**Response:** Thanks. To further study the morphology of generated particles, we carried out Transmission Electron Microscopy (TEM) measurements. Here, we used an atomizer (MSP 1500, MSP) to generate the AS/PA mixture from bulk solution with 46 wt % PA. After selection by DMA1 at dry condition (200 nm at RH below 5 %), the collected samples have been investigated to determine the morphology and mixing state using the TEM. Figure R4 shows aerosol particles are in an initially well-mixed state at dry RH. This suggests that the fast drying "freezes" the mixing state that was present at higher RH in terms of suppressing noticeable liquid-liquid phase separation (LLPS) or crystallization. However, it is difficult to characterize the morphology directly and mixing state of humidified AS/PA aerosol particles at higher RH since these aqueous aerosol particles are sensitive to damage in a strong electron beam under vacuum. A similar phenomenon has been reported in Maskey et al. (2004). They used the term "well-mixed mixture" to represent

that particles are generated from an initially well-mixed solution of AS with organics (e.g., succinic acid and levoglucosan) in order to distinguish the AS/succinic and AS/levoglucosan core-shell mixture, respectively.

[Figure]

**Figure R4.** TEM images of initially well-mixed particles containing AS and 46 wt % PA at RH below 5 %.

For the higher RH range, we make use of the AIOMFAC-LLE model to predict whether and in what range LLPS occurs within AS/PA initially well-mixed aerosol particles containing 46 wt % PA under hydration conditions. As shown in Fig. R5, at low RH, AS is expected to be in a crystalline physical state prior to the deliquescence of AS. Thus, AS is predominantly partitioned to the solid phase (δ), while PA is found in a separate amorphous/liquid phase that further contains water (in solid−liquid equilibrium) up to the complete AS deliquescence at ∼79 % RH. Two liquid phases, one AS-rich (α) and one PA-rich (β), are predicted to coexist between 79 % < RH < 96 % in the hydration case. At RH ∼96 %, the two liquid phases are merging into a single liquid phase, which is predicted as the stable state above 96 % RH. Although, we note that AIOMFAC-LLE may have an uncertainty of several % RH in terms of the RH range in which LLPS is predicted (Song et al., 2012).

In this study, we have systematically investigated the hygroscopicity of AS/PA aerosol particles with different mass fractions of PA in the different mixing states in terms of initial particle

generation. Therefore, we revised "well mixed" as "initially well-mixed" in the response to referee files and the whole paper, including figures. Also, we added the explanation for the term "initially well-mixed aerosol particles". To further avoid the misleading terminality "core-shell", we revised "core-shell" as "core-shell-generated" in the response to referee files and the whole paper, including figures.

[Figure]

**Figure R5.** Predicted equilibrium state phase compositions in mass fractions for aqueous mixtures of AS and PA as a function of water activity (equilibrium RH) at 298 K. Hydration case: a solid−liquid equilibrium is predicted between a solid AS phase (δ; lowest panel) and an aqueous, PA-rich phase (β; middle panel) up to ~96 % RH, followed by liquid−liquid phase separation (coexisting phases α and β) and merging into a single liquid phase at  96 % RH and above.

**Related changes included in the revised manuscript:**

**Page 7 line 144:** We added the explanation for the term "initially well-mixed aerosol particles" here "Due to morphology and mixing state of AS/PA aerosol particles generated by an initially well-mixed aqueous solution as indicated by Fig. S2 at dry RH, note that in the following, aerosol particles generated this way are referred to as initially well-mixed aerosol particles"

**Page 8 line 170:** We added the explanation for the term "core-shell-generated" here "Due to morphology and mixing state of AS/PA aerosol particles generated by a coating-HTDMA at dry RH, note that in the following, aerosol particles generated this way are referred to as core-shell-generated aerosol particles"

**Page 11 line 227:** We add a description about the AIOMFAC-LLE model here "the use of the AIOMFAC-LLE model to predict LLPS and RH-dependent water content of AS/PA particles containing 46 wt % PA under hydration conditions. As shown in Fig. S3. At low RH, AS forms a crystalline phase prior to its deliquescence. Thus, AS is predominantly partitioned to the solid phase (δ), while PA is found in a separate amorphous/liquid phase that further contains water (in solid−liquid equilibrium) up to the complete AS deliquescence at ∼79 % RH. Two liquid phases, one AS-rich (α) and one PA-rich (β), are predicted to coexist between 79 % < RH < 96 % in the hydration case. At RH above 96 %, a single liquid phase is the stable state. Although, we note that AIOMFAC-LLE may have an uncertainty of several % RH in terms of the RH range in which LLPS is predicted (Song et al., 2012)."

*4. Regarding section 2.1.1. As solid crystalline ammonium sulfate is not spherical, how much this affects the estimation of the coating thickness? Further, could the difference between measured GFs and modelled ones be explained by this uncertainty in the coating thickness and further in the calculated organic mass fraction? Even though the coating thickness is calculated to give equivalent mass fraction of PA compared to the well mixed case, I would assume that even small uncertainty in the HTDMA measurements and thus in the coating thickness affects the amount of calculated PA mass fraction. The mass fraction should be marked also to the figure 3.*

**Response:** Thanks for the comment. Zelenyuk et al. (2006) reported that the diameter shape factor of AS increases from 1.03 to 1.07 with increasing mobility diameter from 160 to 500 nm. As you suggested, we calculated the volume equivalent diameter according to the following equation:

$$\chi \frac{d_{ve.dry}}{C_c(K_n(\lambda, d_{ve,dry}))} = \frac{d_{m.dry}}{C_c(K_n(\lambda, d_{m,dry}))} \tag{R1}$$

**Table R2.** 50-nm PA coated with 200-nm AS core

| Core mobility diameter ($d_{m.dry}$ nm) | Shape factor[a] | Volume equilibrium diameter ($d_{ve.dry}$ nm) | Density g/cm$^{-3}$ | PA Coating (nm) | Mass fraction of PA wt %[c] |
|---|---|---|---|---|---|
| 200 | 1 | 200 | 1.77 | 50 | 46 |
| 200 | 1.03 | 196 | 1.65[b] | 54 | 51 |
| 200 | 1.07 | 191 | 1.65[b] | 59 | 55 |

[a]Zelenyuk et al. (2006)

[b]Zelenyuk et al. (2006)

[c]According the equation: $wt_{PA} = \dfrac{\left(\left(\frac{core+coating}{2}\right)^3 - \left(\frac{core}{2}\right)^3\right)\rho_{PA}}{\left(\frac{core}{2}\right)^3 \rho_{AS} + \left(\left(\frac{core+coating}{2}\right)^3 - \left(\frac{core}{2}\right)^3\right)\rho_{PA}}$

Therefore, for the case of 50-nm PA coated with 200-nm AS core aerosol particles, the volume equivalent diameter is 4-nm smaller than the mobility diameter (200 nm) considering the shape factor correction ($\chi$=1.03), which increased by 4 nm in PA thickness and thus higher 5 wt % PA in the 200-nm AS core coated by 50-nm PA. This suggests that slightly large PA loading coating in the core-shell-generated particles. It seems that the non-spherical crystalline AS morphology cannot explain the discrepancy between measured growth factors and the ZSR relation predictions. However, due to the potential presence of  polycrystalline AS, containing pores, cracks, and veins (Zelenyuk et al., 2006; Sjogren et al., 2007), when RH approaches 80 %, these pores or veins may fill with aqueous PA solution. Thus, water molecules may be easier to diffuse to the veins or pores than particle surface. At 80 % RH, deliquesced AS is more likely to mixed partial aqueous solution PA, which may change in morphology of AS/PA core-shell-generated aerosol particles. This is a possible reason for the disagreement between measurements and predictions by the ZSR relation.

**Related changes included in the revised manuscript:**

**Page 12 line 254-line 263:** We revised this sentences "To be specific, for the core-shell-generated aerosol particles consisting of PA and AS, especially at 80 % RH, it shows a considerable amount of water uptake due to the dissolution of the AS core. This dissolution of AS may form completely or partially mixed AS/PA solution droplets. The resulting effect of the arrangement and

restructuring of core-shell-generated structured particles may change the hygroscopicity and mixing state of the core-shell-generated particles at RH above 80 % (Chan et al., 2006; Sjogren et al., 2007). Another morphological effect could be that morphology of a somewhat porous polycrystalline AS core could lead to a larger amount of AS in the particles at RH prior to deliquescence of AS – to appear as a 100-200 nm mobility diameter – hence a thinner than 10-50 PA coating to bring it to a near spherical shape of 110-250 nm core-shell-generated particles (Zelenyuk et al., 2006)." **as**

"To be specific, due to the potential presence of polycrystalline AS, containing pores, cracks, and veins (Zelenyuk et al., 2006; Sjogren et al., 2007), when RH approaches 80 %, these pores or veins may fill with aqueous PA solution. Thus, water molecules may be easier to diffuse to the veins or pores than particle surface. At RH above 80 %, deliquesced AS is more likely to mixed partial aqueous solution PA. The resulting effect of the arrangement and restructuring of core-shell-generated structured particles may change the hygroscopicity, morphology, and mixing state of the core-shell-generated particles (Chan et al., 2006; Sjogren et al., 2007)."

*5. There are numerous English language mistakes, especially in the last half of the paper. I have marked some of them for specific comments below. However, I suggest careful reading and re-writing, especially for the section 4, as there were some paragraphs that I could not understand at all. Related to section 4 (Summary and conclusions), I find that the main results mentioned in the abstract (e.g. at high RH GF decreases as thickness of PA shell increases) and conclusions derived from those are completely missing.*

**Response**: Many thanks, we have carefully revised the whole manuscript regarding language issues, including grammar, wording, and sentence structure, following the reviewer's suggestions; we have rewritten Sect. 4 (Summary and conclusions) they now reads as:

**Page 16 line 345-Page 18 line 381:** "In this study, we focused on PA to represent common organic compounds produced by various sources, (e.g., vehicles, biomass burning, photo-oxidation). It is found that PA aerosol particles uptake water continuously as RH increases. We further investigated the effect of PA coating on the hygroscopicity of core-shell-generated aerosol particles. As PA coating thickness increases, the hygroscopic growth factor of AS/PA core-shell-generated particles increases prior to the deliquescence of AS, but the water uptake decreases at RH above 80 %. Furthermore, we compared the hygroscopic behavior of AS/PA core-shell-generated particles with

that of AS/PA initially well-mixed particles. Due to mixing state effects, higher hygroscopic growth factors of AS/PA core-shell-generated particles, compared to that of initially well-mixed particles, were observed in this study at RH above 80 %. In addition, the ZSR relation prediction is in good agreement with measured results of AS/PA initially well-mixed particles, but leads to underestimation of the hygroscopic growth factor of AS/PA core-shell-generated particles at RH above 80 %. We attribute this discrepancies to the morphology effect when AS deliquesces in the core-shell-generated particles.

There are a vast number of internally mixed organic–inorganic aerosol particles existing in the atmosphere. The hygroscopicity behavior of mixture particles exhibits variability during RH cycles depending on the chemical composition, size, and mixing state. Humidity cycles may lead to liquid–liquid phase separation, e.g. in form of core-shell-generated aerosol particles, including at higher RH or in the salt-supersaturated concentration range. Also, due to the different physicochemical properties of organic compounds (e.g., viscosity, solubility, physical state, and morphology), the equilibrium time varies for these organic coated with inorganic aerosol particles. Therefore, potential kinetic limitations in the HTDMA-measured hygroscopicity of core-shell aerosol particles is to be investigated in both humidification and dehumidification conditions."

*Specific/minor comments:*

*1. The acronym HTDMA is never explained, even though HTDMA instrument/setup is major part of the study.*
**Response**: Many thanks. We revised as:
**Page 4 line 72:** "hygroscopicity tandem differential mobility analyzer (HTDMA)"

*2. Lines 54-56: terms homogeneously mixed and well mixed are both used later in the paper and figures, which is confusing (I understood they mean the same thing). I would select one of them and use it throughout.*
**Response**: Many thanks. We agree with your point. We revised "homogeneously mixed" to "initially well-mixed" in the whole paper. Also, we revised "well mixed" as "initially well-mixed" in the whole paper, including figures.

**Page 3 line 58-61:** "Most of the previous studies on the hygroscopic behavior of multi-component aerosols focus on well-mixed particles generated from initially well-mixed solutions (Miñambres et al., 2010; Shi et al., 2014; Gupta et al., 2015; Jing et al., 2016; Lei et al., 2014; 2018).

**Page 7 line 141:** "**2.1.1 Initially well-mixed AS/PA Aerosol particles**"

**Page 7 line 154:** "**2.1.2 Core-shell-generated AS/PA aerosol particles**"

**Page 9 line 188:** "**3.1 Hygroscopic growth of initially well-mixed aerosol particles**"

**Page 13 line 281-282:** "**3.3 Comparison of core-shell-generated and initially well-mixed AS/PA aerosol particles**"

**Page 14 line 291-293:** "At 75 % RH, the measured growth factor value of core-shell-generated particles is lower than that of initially well-mixed mixtures in the PA mass fraction range from 68 to 46 wt % due to the mass transfer limitations of water vapor transport to the AS core in the core-shell-generated particles."

**Page 2 line 29-30:** "For the AS/PA initially well-mixed particles, a shift of deliquescence relative humidity (DRH) of AS (~80 %, Tang and Munkelwitz (1994)) to lower relative humidity (RH) is observed due to the presence of PA in the initially well-mixed particles."

**Page 3 line 61-64:** "For example, Choi and Chan (2002) studied on the effects of glycerol, succinic acid, malonic acid, citric acid, and glutaric acid on the hygroscopic properties of sodium chloride and AS in the initially well-mixed aerosol particles, respectively, using an electrodynamic balance."

**Page 14 line 288-290:** "However, compared to Fig 5a-b, Fig. 5c shows the hygroscopic growth factors of initially well-mixed AS/PA is slightly higher than that of AS/PA core-shell-generated particles with 46 wt % PA."

**Page 14 line 293-294:** "For the initially well-mixed AS/PA particles, however, partial dissolution of AS into the liquid AP phase may lead to more water uptake by initially well-mixed particles."

**Page 16 line 335-338**: "Next, using the low mass fraction of PA (e.g., 29 wt %) in core-shell-generated aerosol particles is to be investigated."

**Page 31 line 660-662:** "In comparison, the E-AIM model, the fitted expression Eq. (1), and the ZSR relation predicted growth factors of AS, PA, and initially well-mixed particles with different mass fractions of PA, respectively."

[Figure]

 "In comparison to the E-AIM model, the fitted expression Eq. (1), and the ZSR relation predicted growth factors of ammonium sulfate (AS), PA, and initially well-mixed particles with different mass fractions of PA, respectively."

*3. Line 55: …aerosol particles may be divided…*

**Response**: Many thanks.

 We revised as "The initially well-mixed aerosol particles may be divided into homogeneous and heterogeneous internally mixed aerosol particles (Lang-Yona et al., 2009), which could, in turn, strongly influence the water uptake, optical properties, and the cloud condensation nuclei (CCN) ability of the particles (Lesins et al., 2002; Falkovich et al., 2004; Zhang et al., 2005; Schwarz et al., 2006; Su et al., 2010)."

*4. Line 59: …of the earlier studies…*

**Response**: Many thanks.

**Page 3 line 58-61:** We revised as "Most of the previous studies on the hygroscopic behavior of multi-component aerosols focus on well-mixed particles generated from well-mixed solutions (Miñambres et al., 2010; Shi et al., 2014; Gupta et al., 2015; Jing et al., 2016; Lei et al., 2014; 2018)."

*5. Lines 69-71: verb is missing, I assume you mean something like …few laboratory studies have investigated the influence…*

**Response**: Many thanks.

**Page 4 line 69-72:** We revised as "However, to date, few laboratory studies have been investigated on the influence of organic coatings on the hygroscopic behavior of core-shell-generated particles and the difference of mixing state effects on the hygroscopicity of aerosol particles (Zhang et al., 2008; Pagels et al., 2009; Xue et al., 2009; Lang-Yona et al., 2010; Ditas et al., 2018)."

*6. Line 76: change difference to different?*

**Response:** Many thanks.

**Page 4 line 76-77:** We revised as "They suggest that different organic coatings lead to changes in the hygroscopic properties of core-shell-generated particles."

*7. Line 93: correct typo in hygroscopicity*

**Response:** Many thanks.

**Page 5 line 92-94:** We revised as "The organic PA can have profound effect on light scattering, hygroscopicity, and phase transition properties of multicomponent atmospheric aerosols."

*8. Line 104: …aerosol particles have an average…*

**Response**: Many thanks. We didn't revised this sentence because the subject is "occurrence"

*9. Section 2.1: Temperature to what the particles are exposed to is not mentioned, please add it. Ambient temperature was mentioned later at some point, but please be more specific as the experiments should be repeatable by others.*

**Response**: Thanks. We added more information on coating device in Sect. 2.1.

**Page 8 line 161:** We added "The temperature required for vaporizing PA is between ~100 and ~130°C, which corresponds to coating thickness between 10 and 50 nm."

*10. Line 166: …aerosols were pre-humidified…*

**Response:** Many thanks.

**Page 8 line 166-167:** We revised as "After core-shell-generated particle-sizing, aerosols were pre-humidified in a Nafion tube and flowed into the second Nafion humidifier at the set RH2 to reach equilibrium for growth of aerosol particles."

*11. Line 168: I am not sure if I understand what you mean by "Finally the conditioning core-shell particle" here (referring to "the conditioning"). Maybe just the core-shell particle or remaining core-shell particle?*

**Response:** Many thanks.

**Page 8 line 167-168:** We revised as "Finally, the humidified core-shell-generated particles were detected by a DMA3 and a CPC at ambient temperature."

*12. Line 174: Is the equation (1) now from the AIOMFAC model mentioned in lines 117-122? If it is, I would also mention that here as by first reading time I was wondering from where Eq. (1) comes from.*

**Response:** Many thanks. No, the equation (1) is not from AIOMFAC model. Brooks et al. (2004) and Kreidenweis et al. (2014) proposed the equation (1) to predict the continuous hygroscopic growth of aerosol particles, especially for aerosol particles without phase transitions in both hydration and dehydration processes. The detail information has been described in Sect. 2.2.1.

*13. Line 211: …AS in the…particles dissolves…*

**Response:** Many thanks.

**Page 10 line 209-212:** We revised as "For example, in the case of 1:3 mixtures of AS:PA (by mass), 75 wt % PA in the initially well-mixed particles suppresses the deliquescence of AS, i.e., AS in the initially well-mixed particles slowly dissolves into the liquid phase due to continuous water uptake of PA prior to DRH of AS (80 % RH)."

*14. Lines 231 and in some other places: when reporting numbers, I believe the correct way is to use "and" before the last number (i.e. 100, 150 and 200nm instead of 100, 150, 200nm)*

**Response:** Many thanks. I agreed with your point. I checked and revised them in the whole paper.

**Page 8 line 156-158:** We revised as "After a passage through a silica gel diffusion dryer and a neutralizer, the AS core aerosol particles with a certain diameter (100, 150, and 200 nm, respectively) were firstly selected by a DMA1 and then exposed to organic vapors in a coating system."

**Page 9 line 193-195:** We revised as "However, an abrupt increase in the hygroscopic growth factor is observed at 75 % RH for initially well-mixed particles containing 50 and 75 wt % PA, of which the growth factor is higher than that of pure PA aerosol particles ($1.09 \pm 0.01$ nm from measurements shown in Fig. 2) at the same RH."

**Page 10 line 201-204:** We revised as "For example, the measured growth factors for initially well-mixed particles containing 25, 50, and 75 wt % PA are 1.36, 1.28, and 1.19 at 80 % RH, respectively, lower than the growth factor of 1.45 for pure deliquesced AS particles (value from measurements shown in Fig. 2) at the same RH."

**Page 11 line 230-231:** We revised as "Here, we investigated the hygroscopic behavior of samples of various AS core particle sizes (AS particle dry diameter of 100, 150, and 200 nm) and coating (PA coating of 10, 20, 30, and 50 nm), respectively."

**Page 11 line 237-239:** We revised as "For example, the measured growth factor value at 80 % RH is 1.45, 1.40, 1.32, and 1.28 for core-shell-generated particles containing 100 nm AS and 10, 20, 30, and 50 nm coating PA shell, respectively."

**Page 12 line 265-Page 13 line 268:** We revised as "In the case of 50-nm PA shell coated with a certain size of the AS core (100, 150, and 200 nm) with respect to 68, 55, and 46 wt % PA in the core-shell-generated particles, it exhibits an increase in hygroscopic growth factor of core-shell-generated particles at RH below 80 % as the size of AS core decreases."

*15. Line 246: literatures -> literature*

**Response**: Many thanks.

**Page 12 line 246-247:** We revised as "The underprediction of the ZSR relation was also observed in the literature (Chan et al., 2006; Sjogren et al., 2007)."

*16. Line 247: ...observed a strong higher water uptake... -> unclear, possibly "observed higher water uptake"*

**Response**: Many thanks.

**Page 12 line 246-247:** We revised as "Sjogren et al. (2006) observed an enhanced water uptake of mixtures consisting of AS and adipic acid with different mass ratios (1:2, 1:3, and 1:4) at RH above 80 % compared with ZSR relation in the hydration condition."

17. Line 248: correct typo: mass rations -> mass ratios

**Response**: Many thanks.

**Page 12 line 246-247:** We revised as "Sjogren et al. (2006) observed an enhanced water uptake of mixtures consisting of AS and adipic acid with different mass ratios (1:2, 1:3, and 1:4) at RH above 80 % compared with ZSR relation in the hydration condition."

*18. Lines 259-260: very long sentence, hard to follow. Could you reformulate/have two sentences instead?*

**Response:** Many thanks.

**Page 12 line 254-line 263:** We revised this sentences "To be specific, for the core-shell-generated aerosol particles consisting of PA and AS, especially at 80 % RH, it shows a considerable amount of water uptake due to the dissolution of the AS core. This dissolution of AS may form completely or partially mixed AS/PA solution droplets. The resulting effect of the arrangement and restructuring of core-shell-generated structured particles may change the hygroscopicity and mixing state of the core-shell-generated particles at RH above 80 % (Chan et al., 2006; Sjogren et al., 2007). Another morphological effect could be that morphology of a somewhat porous polycrystalline AS core could lead to a larger amount of AS in the particles at RH prior to deliquescence of AS – to appear as a 100-200 nm mobility diameter – hence a thinner than 10-50 PA coating to bring it to a near spherical shape of 110-250 nm core-shell-generated particles (Zelenyuk et al., 2006)." **as**

"To be specific, due to the potential presence of  polycrystalline AS, containing pores, cracks, and veins (Zelenyuk et al., 2006; Sjogren et al., 2007), when RH approaches 80 %, these pores or veins may fill with aqueous PA solution. Thus, water molecules may be easier to diffuse to the veins or pores than particle surface. At RH above 80 %, deliquesced AS is more likely to mixed partial aqueous solution PA. The resulting effect of the arrangement and restructuring of core-shell-generated structured particles may change the hygroscopicity, morphology, and mixing state of the core-shell-generated particles (Chan et al., 2006; Sjogren et al., 2007)."

*19. Line 276: change "it assumes" into "it is assumed that"*

**Response**: Many thanks.

**Page 13 line 276-278:** We revised as "For ZSR prediction, it is assumed that volume fraction of AS components is constant according to the ratio of volume of AS core in the sphere to the volume of core-shell-generated sphere based on Eq. (3)."

*20. Line 276: please remind the reader about the morphology effect also here with a short sentence in addition to referring section 3.2.*

**Response**: Many thanks.

**Page 13 line 274-276:** We added some sentences.

"The discrepancy between measured hygroscopic growth factors and predicted hygroscopic growth factors of core-shell-generated particles by ZSR relation, as discussed in Sect. 3.2 above, is due to the morphology effect. The porous and polycrystalline AS core deliquesces at 80 % RH, and aqueous PA solutions are filled with these veins and cavities of water-soluble AS, which leads to change in mixing state/morphology. This affecting hygroscopicity of core-shell-generated aerosol particles at RH above 80 %."

*21. Line 289: correct "is" to "are" as you have plural*

**Response**: Many thanks.

**Page 14 line 288-290:** We revised as "However, compared to Fig 5a-b, Fig. 5c shows the hygroscopic growth factors of initially well-mixed AS/PA are slightly higher than that of AS/PA core-shell-generated particles with 46 wt % PA."

*22. Line 294: AP most likely typo, please correct*

**Response**: Many thanks.

**Page 14 line 293-294:** We revised as "For the initially well-mixed AS/PA particles, however, partial dissolution of AS into the liquid PA phase may lead to more water uptake by initially well-mixed particles."

*23. Line 298-299: The sentence is unclear, maybe remove "the"?*

**Response**: Many thanks.

**Page 14 line 293-294:** We revised "Core-shell-generated particle morphology may experience the restructuring and associate size change of particles." as

"Accordingly, at high RH, the occurrence of microscopic restructuring of core-shell-generated particles may affect their size."

*24. Lines 301-304: very long sentence, so what was actually observed?*

**Response**: Many thanks.

**Page 14 line 301-304:** We revised as "Chan et al. (2006) investigated hygroscopicity of 49 wt % glutaric acid coated on AS core during two continuous hydration cycles. They observed that the experimental growth factor of the fresh core-shell of AS and glutaric acid in the first hydration cycle is a bit higher than those in second hydration cycle with the same mass fractions of glutaric acid."

*25. Line 309: repetition (observation was observed). Maybe change to ...a contracting observation was made...*

**Response**: Many thanks.

**Page 14 line 309-Page 15 line 310:** We revised as "However, a contrasting phenomenon was observed in the previous study (Maskey et al., 2014)."

*26. Line 332-333: I do not understand the sentence starting "they found that...", please elaborate/correct sentence structure*

**Response**: Many thanks.

**Page 14 line 332-334:** We revised as "they found that the slightly higher growth factors of the initially well-mixed particles is than that of core-shell-generated aerosol particles (12-nm levoglucosan coated 88-nm AS)."

*27. Lines 338-342: Again, very long and complicated sentence, hard to follow. Please simplify."*

**Page 16 line 338-342:** We revised as "In addition, for the AS/PA mixture aerosol particles containing 46-68 wt % PA, the measured growth factors of initially well-mixed AS/PA particles are in good agreement with the ZSR relation prediction comparing with that of core-shell-generated AS/PA particles."

*28-36 comments as follows:*

*Line 352: change "estimation of" into "estimator for"*

*Lines 357-359: It seems there is main verb missing?*

*Lines 360-361: "According to filed studies…a variety of organic aerosol particles were characterised in the atmosphere…" This seems a little off from the context. Or do you mean that also in other studies various particle properties have been characterised? By first reading it seems you are writing about identifying particles, which is not related to your study in that sense.*

*Lines 366-367: …to depend on the difference of influence of kinetic limitations". Please reduce the use of "of" for clarity.*

*Line 369: change "significant" to "important"?*

*Line 370: change "combining" to "combined"*

*Line 371: change "organics coating" to "organic coatings"*

*Line 370-375: starting from "understanding…". Very long sentence, please break into smaller sections & elaborate, as now it is really hard to understand*

*Line 377: are the humidity cycles or particles depending on the ambient RH history? Make this clear in the text.*

**Response**: Many thanks, we have carefully revised the whole manuscript regarding language issues, including grammar, wording, and sentence structure, following the reviewer's suggestions, we have rewritten Sect. 4 (Summary and conclusion) they now reads as:

**Page 16 line 345-Page 18 line 381:** "In this study, we focused on PA to represent common organic compounds produced by various sources, (e.g., vehicles, biomass burning, photo-oxidation). It is found that PA aerosol particles uptake water continuously as RH increases. We further investigate the effect of PA coating on the hygroscopicity of core-shell-generated aerosol particles. As PA coating thickness increases, the hygroscopic growth factor of AS/PA core-shell-generated increases prior to the deliquescence of AS, while the water uptake decreases at RH above 80 %. Furthermore, we compared the hygroscopic behavior of AS/PA core-shell-generated particles with that of AS/PA initially well-mixed particles. Due to the mixing state effects, the higher hygroscopic growth factor of AS/PA core-shell-generated than that of initially well-mixed particles was observed in this study at RH above 80 %. In addition, the ZSR relation prediction is in good agreement with measured results of AS/PA initially well-mixed particles, but underestimation of

the hygroscopic growth factor of AS/PA core-shell-generated particles at RH above 80 %. We attribute to the morphology effect when AS deliquesces in the core-shell-generated particles.

As we know, there are a vast number of mixture aerosol particles existing in the atmosphere. Their hygroscopic behavior of mixture particles exhibits variability during RH cycle depending on the chemical composition, size, and mixing state. Humidity cycles may lead to liquid-liquid phase separation of core-shell aerosol particles at higher RH or supersaturated concentration range. Also, due to the different physicochemical properties of organic compounds (e.g., viscosity, solubility, physical state, and morphology), the equilibrium time varies for these organic coated with inorganic aerosol particles. Therefore, the kinetic limitation of hygroscopicity of core-shell aerosol particles is to be investigated in both humidification and dehumidification conditions."

**Reference:**

Brooks, S. D., DeMott, P. J., and Kreidenweis, S. M.: Water uptake by particles containing humic materials and mixtures of humic materials with ammonium sulfate, Atmospheric Environment, 38, 1859-1868, 2004.

Chan, M. N., Lee, A. K. Y., and Chan, C. K.: Responses of ammonium sulfate particles coated with glutaric acid to cyclic changes in relative humidity: Hygroscopicity and Raman characterization, Environmental Science & Technology, 40, 6983-6989, 2006.

Ciobanu, V. G., Marcolli, C., Krieger, U. K., Weers, U., and Peter, T.: Liquid-liquid phase separation in mixed organic/inorganic aerosol particles, J. Phys. Chem., 113 10966–10978, 2009.

Gupta, D., Kim, H., Park, G., Li, X., Eom, H. J., and Ro, C. U.: Hygroscopic properties of NaCl and $NaNO_3$ mixture particles as reacted inorganic sea-salt aerosol surrogates, Atmos. Chem. Phys., 15, 3379-3393, 2015.

Hameri, K., Charlson, R., and Hansson, H. C.: Hygroscopic properties of mixed ammonium sulfate and carboxylic acids particles, Aiche Journal, 48, 1309-1316, 2002.

Henning, S., Ziese, M., Kiselev, A., Saathoff, H., Möhler, O., Mentel, T. F., Buchholz, A., Spindler, C., Michaud, V., Monier, M., Sellegri, K., and Stratmann, F.: Hygroscopic growth and droplet activation of soot particles: uncoated, succinic or sulfuric acid coated, Atmos. Chem. Phys., 12, 4525-4537, 2012.

Hodas, N., Zuend, A., Mui, W., Flagan, R. C., and Seinfeld, J. H.: Influence of particle-phase state on the hygroscopic behavior of mixed organic–inorganic aerosols, Atmos. Chem. Phys., 15, 5027-5045, https://doi.org/10.5194/acp-15-5027-2015, 2015.

Jing, B., Tong, S., Liu, Q., Li, K., Wang, W., Zhang, Y., and Ge, M.: Hygroscopic behavior of multicomponent organic aerosols and their internal mixtures with ammonium sulfate, Atmospheric Chemistry and Physics, 16, 4101-4118, 2016.

Kleindienst, T. E., Smith, D. F., Li, W., Edney, E. O., Driscoll, D. J., Speer, R. E., and Weathers, W. S.: Secondary organic aerosol formation from the oxidation of aromatic hydrocarbons in the presence of dry submicron ammonium sulfate aerosol, Atmos Environ, 33, 3669-3681, 1999.

Lei, T., Zuend, A., Cheng, Y., Su, H., Wang, W., and Ge, M.: Hygroscopicity of organic surrogate compounds from biomass burning and their effect on the efflorescence of ammonium sulfate in mixed aerosol particles, Atmos. Chem. Phys., 18, 1045-1064, 2018.

Lei, T., Zuend, A., Wang, W. G., Zhang, Y. H., and Ge, M. F.: Hygroscopicity of organic compounds from biomass burning and their influence on the water uptake of mixed organic ammonium sulfate aerosols, Atmospheric Chemistry and Physics, 14, 1-20, 2014a.

Lei, T., Zuend, A., Wang, W. G., Zhang, Y. H., and Ge, M. F.: Hygroscopicity of organic compounds from biomass burning and their influence on the water uptake of mixed organic ammonium sulfate aerosols, Atmos. Chem. Phys., 14, 11165-11183, 2014b.

Maskey, S., Chong, K. Y., Kim, G., Kim, J.-S., Ali, A., and Park, K.: Effect of mixing structure on the hygroscopic behavior of ultrafine ammonium sulfate particles mixed with succinic acid and levoglucosan, Particuology, 13, 27-34, 2014.

Mikhailov, E. F., Vlasenko, S. S., and Ryshkevich, T. I.: Influence of chemical composition and microstructure on the hygroscopic growth of pyrogenic aerosol, Izvestiya, Atmospheric and Oceanic Physics, 44, 416-431, 2008.

Miñambres, L., Sánchez, M. N., Castaño, F., and Basterretxea, F. J.: Hygroscopic Properties of Internally Mixed Particles of Ammonium Sulfate and Succinic Acid Studied by Infrared Spectroscopy, The Journal of Physical Chemistry A, 114, 6124-6130, 2010.

Mochida, M. and Kawamura, K.: Hygroscopic properties of levoglucosan and related organic compounds characteristic to biomass burning aerosol particles, Journal of Geophysical Research-Atmospheres, 109, 2004.

Peng, C., Chan, M. N., and Chan, C. K.: The hygroscopic properties of dicarboxylic and multifunctional acids: Measurements and UNIFAC predictions, Environmental Science & Technology, 35, 4495-4501, 2001.

Shi, Y. J., Ge, M. F., and Wang, W. G.: Hygroscopicity of internally mixed aerosol particles containing benzoic acid and inorganic salts, Atmospheric Environment, 60, 9-17, 2012.

Shiraiwa, M., Zuend, A., Bertram, A. K., and Seinfeld, J. H.: Gas-particle partitioning of atmospheric aerosols: interplay of physical stte, non-ideal mixing, and morphology, Phys. Chem. Chem Phys., 15, 11441-11453, 2013.

Sjogren, S., Gysel, M., Weingartner, E., Baltensperger, U., Cubison, M. J., Coe, H., Zardini, A. A., Marcolli, C., Krieger, U. K., and Peter, T.: Hygroscopic growth and water uptake kinetics of two-phase aerosol particles consisting of ammonium sulfate, adipic and humic acid mixtures, Journal of Aerosol Science, 38, 157-171, 2007.

Song, M., Marcolli, C., Krieger, U. K., Zuend, A., and Peter, T.: Liquid-liquid phase separation and morphology of internally mixed dicarboxylic acids/ammonium sulfate/water particles, Atmos. Chem. Phys., 12, 2691-2712, https://doi.org/10.5194/acp-12-2691-2012, 2012.

Song, M., Ham, S., Andrews, R. J., You, Y., and Bertram, A. K.: Liquid–liquid phase separation in organic particles containing one and two organic species: importance of the average $O : C$, Atmos. Chem. Phys., 18, 12075-12084, https://doi.org/10.5194/acp-18-12075-2018, 2018.

Varutbangkul, V., Brechtel, F. J., Bahreini, R., Ng, N. L., Keywood, M. D., Kroll, J. H., Flagan, R. C., Seinfeld, J. H., Lee, A., and Goldstein, A. H.: Hygroscopicity of secondary organic aerosols formed by oxidation of cycloalkenes, monoterpenes, sesquiterpenes, and related compounds, Atmos. Chem. Phys., 6, 2367-2388, 2006.

Wang, G., Kawamura, K., Xie, M., Hu, S., Li, J., Zhou, B., Cao, J., and An, Z.: Selected water-soluble organic compounds found in size-resolved aerosols collected from urban, mountain and marine atmospheres over East Asia, Tellus B, 63, 371-381, 2011.

Wex, H., Ziese, M., Kiselev, A., Henning, S., and Stratmann, F.: Deliquescence and hygroscopic growth of succinic acid particles measured with LACIS, Geophysical Research Letters, 34, 2007.

Zelenyuk, A., Cai, Y., and Imre, D.: From Agglomerates of Spheres to Irregularly Shaped Particles: Determination of Dynamic Shape Factors from Measurements of Mobility and Vacuum Aerodynamic Diameters, Aerosol Science and Technology, 40, 197-217, 2006.

Wise, M. E., Surratt, J. D., Curtis, D. B., Shilling, J. E., and Tolbert, M. A.: Hygroscopic growth of ammonium sulfate/dicarboxylic acids, J. Geophys. Res., 108(20), 4638, doi:10.1029/2003JD003775, 2003.

---

## Referee Report (RR1)

**Referee comments for "Effect of mixing structure on the water uptake of mixtures of ammonium sulfate and phthalic acid particles" (revised manuscript)**

The revised manuscript shows significant improvements to the quality of the manuscript when compared to the original version. All referee comments are adequately assessed, and especially the uncertainty associated with the mixing state after humidification of the AS/PA particles is justified sufficiently with the added text in the manuscript and figures in SI material. The language of the paper is now also more fluent and better structured to fit the high standards of ACP. I recommend the publication of this paper in ACP as it is now.